# MYSM1 acts as a novel co-activator of ERα to confer antiestrogen resistance in breast cancer

Ruina Luan[1], Mingcong He [ID][1], Hao Li[1], Yu Bai[1], Anqi Wang[1,2], Ge Sun [ID][1], Baosheng Zhou [ID][1], Manlin Wang[1], Chunyu Wang[1], Shengli Wang[1], Kai Zeng[1], Jianwei Feng[1], Lin Lin[1], Yuntao Wei [ID][3], Shigeaki Kato [ID][4,5], Qiang Zhang [ID][3✉] & Yue Zhao [ID][1✉]

## Abstract

Endocrine resistance is a crucial challenge in estrogen receptor alpha (ERα)-positive breast cancer (BCa). Aberrant alteration in modulation of E2/ERα signaling pathway has emerged as the putative contributor for endocrine resistance in BCa. Herein, we demonstrate that MYSM1 as a deubiquitinase participates in modulating ERα action via histone and non-histone deubiquitination. MYSM1 is involved in maintenance of ERα stability via ERα deubiquitination. MYSM1 regulates relevant histone modifications on cis regulatory elements of ERα-regulated genes, facilitating chromatin decondensation. MYSM1 is highly expressed in clinical BCa samples. MYSM1 depletion attenuates BCa-derived cell growth in xenograft models and increases the sensitivity of antiestrogen agents in BCa cells. A virtual screen shows that the small molecule Imatinib could potentially interact with catalytic MPN domain of MYSM1 to inhibit BCa cell growth via MYSM1-ERα axis. These findings clarify the molecular mechanism of MYSM1 as an epigenetic modifier in regulation of ERα action and provide a potential therapeutic target for endocrine resistance in BCa.

**Keywords** Breast Cancer; Epigenetic Modifier; Estrogen Receptor α; MYSM1; Protein Deubiquitination
**Subject Category** Cancer; Chromatin, Transcription & Genomics

## Introduction

Breast cancer has been ranked as the malignancy with the highest incidence worldwide since 2020 and owns the highest mortality among cancers in women, which directly impacts their life quality and expectancy (Sung et al, 2021). Nearly, 70–80% breast cancer is characterized by ERα positive expression (Waks and Winer, 2019). ERα participates in vital cellular processes, such as proliferation, differentiation, and apoptosis of mammary epithelial cells. Given the pleiotropic functions of ERα, perturbation in the estrogen (17β-estradiol, E2)/ERα signaling pathway could result in BCa initiation and progression. The causal role of ERα in BCa pathology makes it a predictive factor and therapeutic target of BCa. At present, endocrine therapy blocking ERα pathway is an exactly prevailing treatment for ERα-positive BCa (Mehta et al, 2019). While most cases are originally sensitive to antiestrogen therapies, the adaptability of tumor cells leads to a substantial percentage of patients stopping responding and gradually developing drug resistance (Hanker et al, 2020). Thus, well understanding the mechanism underlying the modulation of E2/ERα signaling pathway would provide the potential strategies for endocrine resistance.

ERα belongs to a member of the steroid hormone receptor superfamily. E2 binding activates ERα by changing its conformation, thereby transferring into the nucleus to induce the transcription of ERα target genes (Hewitt and Korach, 2018; Yasar et al, 2017). Basic transcriptional machinery along with co-regulators are rapidly recruited to modulate the expression of target genes, such as c-Myc, CCND1, GREB1, TFF1 (Metivier et al, 2003; Shang et al, 2000). The complicated network of these co-regulators defines a code, which acts as an adapter molecule connecting ERα to the basal transcription apparatus or alters chromosome structure on cis-regulatory elements, comprising covalent modifications in histone tails and nucleosome remodeling (Dimitrakopoulos et al, 2021). Accumulating evidence suggests that the core regulatory proteins of ERα action subtly tune hormone sensitivity, receptor stability, and ERα-mediated transcriptional activity according to their diverse enzymatic activities and associated patterns (Manavathi et al, 2013; Shao et al, 2004; Sukocheva et al, 2020). A series of ERα cofactors have been identified so far to regulate the estrogen-driven transcriptional program. CBP/p300, p/CAF, the p160 family, PELP1, SWI/SNF complex, YAP, and PARP-1 identified as ERα co-activators lead to chromatin de-condensation and modulate epigenetic changes, providing a selective advantage for cancer cell growth, differentiation, invasion, metastasis, and endocrine resistance (Gadad et al, 2021; Garcia-Martinez et al, 2021; Ju et al, 2006; Schiewer and Knudsen, 2014; Zhu et al, 2019). While the co-repressors of ERα, such as SMRT, NCOR1, PIP140, BRCA1, MTA1, TLE3, play multiple functions on breast cancer processes through

[1]Department of Cell Biology, Key Laboratory of Medical Cell Biology, Ministry of Education, School of Life Sciences, China Medical University, 110122 Shenyang City, Liaoning Province, China. [2]First Clinical Medical College, China Medical University, 110001 Shenyang City, Liaoning Province, China. [3]Department of Breast Surgery, Cancer Hospital of China Medical University, Liaoning Cancer Hospital & Institute, 110042 Shenyang City, Liaoning Province, China. [4]Graduate School of Life Science and Engineering, Iryo Sosei University, Iino, Chuo-dai, Iwaki, Fukushima 9708551, Japan. [5]Research Institute of Innovative Medicine, Tokiwa Foundation, Iwaki, Fukushima, Japan.
✉E-mail: zhangqiang8220@163.com; yzhao30@cmu.edu.cn

modulation of ERα action (Dobrzycka et al, 2003; Wen et al, 2009). It is convictive that accurate orchestration of ERα action results from the coordination of multiple co-regulators. Since ER conducts a diverse function, identification of novel ERα co-regulator is still necessary for finding the potential target for ERα-positive breast cancer treatment.

Myb like, SWIRM and MPN domains 1 (MYSM1) is a metalloproteinase with deubiquitinase catalytic activity. It enhances chromatin accessibility by specifically removing H2Aub to facilitate gene transcription (Zhu et al, 2007). MYSM1 is involved in extensive physiological processes. In the hematopoietic system, MYSM1 de-represses an array of genes which are pivotal for lineage specification and stem cell differentiation through histone H2A deubiquitination (Belle et al, 2020; Nijnik et al, 2012; Wang et al, 2013). Additionally, MYSM1 guards against excessive inflammation and autoimmune reaction under circumstances of inflammatory irritation or infection by non-histone deubiquitination to abrogate NOD2:RIP2, cGAS-STING, or TRAF3/6 signaling in the immune system (Panda and Gekara, 2018, Panda et al, 2015; Tian et al, 2020b). MYSM1 deficiency spontaneously perturbs the proper repair of DNA damage to accumulate DNA double strand breaks (DSB), accompanied by cellular senescence (Kroeger et al, 2020; Nishi et al, 2014; Tian et al, 2020a). MYSM1 is also involved in tumor pathologic processes. It has been reported that MYSM1 suppresses colorectal cancer progression via histone H2A deubiquitination, thereby activating miR-200 family members/CDH1 (Chen et al, 2021). MYSM1 co-activates androgen receptor (AR) action to promote prostate cancer (Zhu et al, 2007), while MYSM1 inhibits castration-resistant prostate cancer (CRPC) process through PI3K/Akt signaling regulation (Sun et al, 2019). In triple-negative breast cancer (TNBC), MYSM1 reduces RSK3 protein to inactive RSK3-phospho-BAD pathway and induces apoptosis and cisplatin sensitivity (Guan et al, 2022). However, the molecular mechanisms underlying the modulation function of MYSM1 on ERα signaling in ERα-positive BCa remains elusive.

In this study, we identify MYSM1 as a co-activator of ERα in a *Drosophila* model carrying an ERα-mediated gene transcription system. We further demonstrate that MYSM1 associates with ERα and increases ERα-induced transcriptional activity through non-histone and histone deubiquitination in breast cancer-derived cell lines. Unexpectedly, our data suggests that MYSM1 stabilizes the ERα protein on Lysine 48 (K48) and K63-linked poly-ubiquitination via its deubiquitinase activity. On the other hand, MYSM1 is required for the association between ERα and p300/CBP/pCAF, the core components of a classical histone acetyltransferases (HATs) complex. MYSM1 is recruited together with ERα to the promoter region of E2-induced genes, leading to epigenetic modulation of H2Aub, H3K4me3, H3K9ac and H3K27ac levels. MYSM1 depletion suppresses cell proliferation and increases antiestrogen sensitivity in BCa-derived cell lines. By screening the existing compounds in the ZINC database, we have identified Imatinib as a small molecule that suppresses the enhancement of MYSM1 on ERα-mediated transactivation via inhibition of the catalytic activity of MYSM1. Moreover, Imatinib suppresses cell growth and enhances the sensitivity of Tamoxifen-insensitive cells to Tamoxifen treatment via the MYSM1-ERα pathway. Furthermore, MYSM1 is highly expressed in clinical BCa samples, and higher expression of MYSM1 predicts a poor survival of BCa patients. Taken together, our findings suggest that the deubiquitinase activity

of MYSM1, in addition to its epigenetic modulation function as a co-activator of ERα, is involved in non-histone modification to maintain ERα stability in ERα-positive BCa progression.

# Results

## MYSM1 physically associates with ERα in breast cancer cells

We have previously identified several co-regulators of AR in *Drosophila* experimental system (Sun et al, 2016; Wang et al, 2015). Herein, we generated an inducible *Drosophila* model containing an ERα-mediated gene transcription system to isolate co-regulators involved in modulating ERα actions. The system includes a GAL4 expression construct, a UAS-linked ERα expression construct, an ERE-linked green fluorescent protein (GFP) reporter construct with the modified GAL4-UAS bipartite approach in *Drosophila* (Fig. 1A). ERα-induced transactivation can be monitored by the intensity of GFP expression in *Drosophila* experimental system. The experimental *drosophila* model was crossed with *Drosophila* mutants of potential genes, and GFP protein expression was assessed to screen ERα cofactors in the presence of E2. Interestingly, loss of function mutant of *CG4751* dramatically compromised ERα-induced GFP protein expression. To confirm this result, we further generated transgenic *Drosophila* with human homolog of *CG4751* (*MYSM1*), the results demonstrated that MYSM1 increased ERα-mediated transactivation, suggesting that MYSM1 may be a co-activator of ERα (Fig. EV1A).

We thus turn to ask whether MYSM1 associates with ERα. STRING database (http://www.bork.embl-heidelberg.de/STRING/) predicted that MYSM1 may correlate with ERα (Fig. EV1B). Then, we set out to examine the interaction between MYSM1 and ERα in mammalian cells by Co-immunoprecipitation (Co-IP). ERα and FLAG-tagged MYSM1 expression plasmids were co-transfected into HEK293 cells. And specific antibodies against FLAG was used to precipitate the related protein as indicated. Our results showed that MYSM1 associated with ERα (Fig. 1B). The similar experiments were performed in ERα-positive breast cancer (BCa)-derived cell lines (MCF-7 and T47D). The results showed that the endogenous MYSM1 associated with ERα (Figs. 1C,D and EV1C,D). To determine the exact interaction region in MYSM1, we constructed MYSM1 truncated mutant plasmids as indicated (Fig. 1E). Co-IP experiments were performed with co-transfection of these plasmids and ERα expression plasmid in mammalian cells. The precipitation results demonstrated that loss of SANT (Swi3, Ada2, N-CoR, (TF) IIIB) domain more obviously impaired the association between MYSM1 and ERα, compared with that of SWIRM or MPN domain (Figs. 1F and EV1E). Additionally, GST-pull down was performed with GST-tagged ERα-AF1 (29–180aa) or ERα-AF2 (282–595aa) fragments (Zeng et al, 2020). The results showed that MYSM1 directly interacted with ERα-AF2 fragment (Fig. 1G).

To investigate the subcellular distribution of MYSM1 and ERα, we applied immunofluorescence (IF) in HEK293 cells and breast cancer cells (MCF-7 and T47D) (Figs. 1H,I and EV1F). We observed that MYSM1 is distributed in the nucleus regardless of E2 ($10^{-7}$ M) treatment, whereas ERα diffused into the nucleus and entirely co-located with MYSM1 in the presence of E2. HEK293

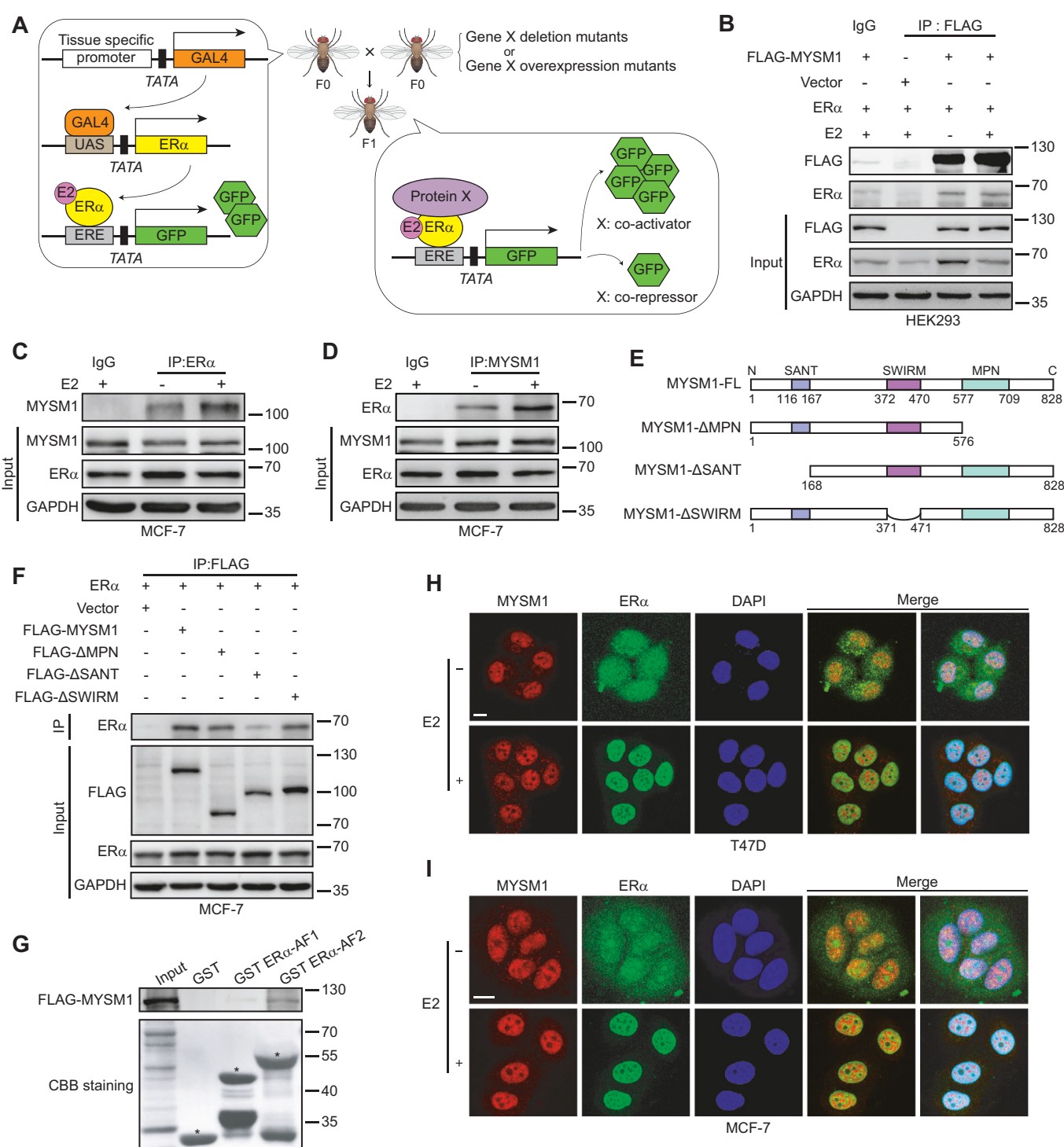

cells were transfected with MYSM1 truncated mutant plasmids and ERα expression plasmid. The results from IF experiments demonstrated that three kinds of mutants of MYSM1 (deletion of MPN, SWIRM, or SANT) were distributed in the nucleus, indicating that nuclear localization signal (NLS) of MYSM1 may exist in other regions except MPN, SWIRM, and SANT domains (Fig. EV1F). Collectively, these data suggest that MYSM1 interacts with ERα.

## MYSM1 co-activates ERα-induced transcriptional activity

Having established that ERα-mediated transactivation is significantly down-regulated by loss of function of CG4751 (*Drosophila* homolog of MYSM1) in *Drosophila* experimental system, we speculated that MYSM1 may functionally participates in ERα signaling pathway in human. Luciferase assays was then performed to examine the effect of MYSM1 on modulation of ERα action. The

◄ **Figure 1. MYSM1 interacts with ERα in breast cancer cells.**

(A) Schematic diagram of the co-regulator screening *Drosophila* model carrying an ERα-mediated gene transcription system. The F0 parental generation with a GAL4-UAS driver targeted ERα expression and an ERE-inserted GFP reporter was crossed with species harboring particular gene X deletion or overexpression mutants. GFP expression changes in the eye discs of F1 progenies were examined to appraise the effects of gene X mutants on ERα-induced transactivation with E2 treatment. (B) Co-immunoprecipitation showing the interaction between exogenous MYSM1 and ERα with or without E2. HEK293 cells were co-transfected with FLAG-tagged MYSM1 and ERα plasmids and incubated in estrogen-depleted medium (5% charcoal-stripped serum in phenol red-free DMEM) for 48 h, then treated with ethanol vehicle or E2 (100 nM) for 12 h. Whole-cell extracts were harvested for immunoprecipitation with the anti-FLAG or IgG antibodies, with 5% reserved as the input control. (C, D) Reciprocal Co-immunoprecipitation to detect the association between endogenous MYSM1 and ERα in MCF-7 cells with the indicated antibodies. (E) Schematic diagram representing the MYSM1 full length and truncated expression plasmids (ΔMPN, ΔSANT, ΔSWIRM) containing FLAG tags. (F) Co-immunoprecipitation to determine the exact domains in MYSM1 responsible for its binding to ERα. Ectopic ERα along with wild-type MYSM1 or MYSM1 deletion mutants were expressed in MCF-7 cells. Anti-FLAG-MYSM1 immunoprecipitates or corresponding input were then immunoblotted for ERα, MYSM1 (FLAG) and GAPDH. (G) Detection of the directly binding of MYSM1 and ERα by GST-pull down assay. GST, GST ERα-AF1, GST ERα-AF2 proteins expressed in an *E. coli* system were incubated with the in vitro transcribed 35S-MYSM1 protein. The binding proteins in the reaction mixtures were analyzed by SDS-PAGE and autoradiography. Asterisks marked GST fusion protein locations. (H, I) Immunofluorescence showing the co-localization of MYSM1 (red) and ERα (green) in T47D (H) and MCF-7 (I) cells. Nuclei were stained with DAPI (blue). Scale bars, 10 μm. Data information: Results in (B–D, F–I) are representatives of three independent experiments performed in duplicate. Source data are available online for this figure.

results demonstrated that MYSM1 co-activated ERα-induced transactivation in a dose-dependent manner in HEK293 cells (Figs. 2A and EV2A). Meanwhile, MYSM1 knockdown inhibits ERα action in T47D cells, indicating that MYSM1 is a novel co-activator of ERα (Fig. EV2D). ERα mainly contains N-terminal activation function-1 (ERα-AF1) carrying constitutive ligand-independent domain, and C-terminal activation function-2 (ERα-AF2) harboring ligand-dependent domain. We sought to examine the effect of MYSM1 on transactivation induced by the ERα truncated mutants, including ERα-AF1 (1-182aa) and ERα-AF2 (178-595aa) as indicated (Fig. EV2B). The results showed that MYSM1 simultaneously increased ERα-AF1 and AF2 actions (Figs. 2B and EV2C). To gain insight into the functional domain of MYSM1, we further performed the similar experiments with MYSM1 truncated mutant plasmids. Compared with the full length of MYSM1, the lack of SWIRM domain had little effect on ERα-induced transactivation, while MYSM1-ΔSANT or MYSM1-ΔMPN dramatically reduced ERα action (Fig. 2C), indicating that the SANT or MPN domain is required for the co-activation function of MYSM1 on ERα-mediated transactivation.

Indeed, ERα behaves in numerous biological processes through its target genes. To verify the effect of MYSM1 on regulation of E2/ERα signaling pathway, quantitative real-time PCR (qPCR) experiments were performed to examine co-activation function of MYSM1 on ERα target gene transcription in ERα-positive breast cancer cells. The data demonstrated MYSM1 knockdown significantly suppressed the transcription of endogenous ERα regulated genes, including *c-Myc*, *CCND1*, *VEGF*, *GREB1*, *TFF1*, *E2F1*, *FOXC1* (Figs. 2D and EV2E). Western blot was further conducted for examining the regulation function of MYSM1 on ERα action. The results showed that MYSM1 depletion inhibited the protein levels of ERα regulated genes, such as c-Myc, CCND1, and VEGF (Figs. 2E and EV2F). Consistent with this, ectopic expression of MYSM1 increased c-Myc, CCND1 and VEGF protein level (Figs. 2F and EV2G). Moreover, additional transfection with ERα expression plasmids could reverse the reduced expression of ERα target genes caused by siRNA against MYSM1 (siMYSM1) in breast cancer cells (Figs. 2G and EV2H). A rescue studies were further performed in BCa cells carrying the wild-type or MPN domain-deletion MYSM1 expression plasmid together with siMYSM1 as indicated. The results showed that the catalytically mutant MYSM1 hardly changed ERα target gene expression, suggesting MYSM1 enzyme activity is required for up-regulation of ERα action (Figs. 2H and

EV2I). Taken together, our results suggest that MYSM1 acts as a novel co-activator of ERα.

## MYSM1 stabilizes the ERα protein with its deubiquitination activity

Unexpectedly, in western blot experiments as shown in Fig. 2E,F, we observed that ectopic expression of MYSM1 increased ERα protein expression, while knockdown of MYSM1 reduced ERα protein level. This prompted us to ask how MYSM1 influences ERα protein expression in breast cancer. shRNA lentivirus against MYSM1 (shMYSM1) was infected into ER-positive breast cancer cell lines. The results showed that knockdown of MYSM1 decreased ERα protein level, whereas ERα mRNA expression had no significant change (Fig. 3A,B). On the other hand, ectopic expression of MYSM1enhanced ERα expression in MCF-7 and HEK293 (Fig. 3D), indicating that MYSM1 up-regulates ERα protein itself at the post-transcriptional level. When cycloheximide (CHX) was applied at specified time points to inhibit protein synthesis, depletion of MYSM1 significantly accelerated ERα degradation and ectopic expression of wild-type MYSM1 plasmid (MYSM1-FL) ameliorated ERα degradation process (Fig. 3C,E). In addition, we treated MCF-7 cells with proteasome inhibitor (MG132), the results showed that regulation of MYSM1 on ERα protein expression was prevented by MG132 treatment (Fig. 3F,G), suggesting that MYSM1 is involved in maintenance of ERα protein stability.

MYSM1 belongs to the JAMM family of deubiquitinase, so we hypothesized it may participate in maintenance of ERα stability through its deubiquitinase activity. Next, we turned to perform western blot to detect the effect of full length of MYSM1 (MYSM1-FL) and catalytically loss of function of MYSM1 mutant (MYSM1-ΔMPN) on ERα protein expression. The results demonstrated MYSM1-ΔMPN largely decreased the enhancement of ERα expression induced by MYSM1-FL (Fig. 3D,E), indicating that deubiquitinase activity of MYSM1 is indispensable for maintenance of ERα stabilization. Ubiquitination assays based on immunoprecipitation were further performed to determine whether ERα is a substrate of MYSM1. Polyubiquitinated ERα proteins were visibly enriched by immunomagnetic beads in the control group, and MYSM1 depletion significantly enhanced ubiquitination of ERα (Fig. 3H). In addition, the level of ERα ubiquitination underwent a sharp decline by MYSM1, while seemed to remain constant with

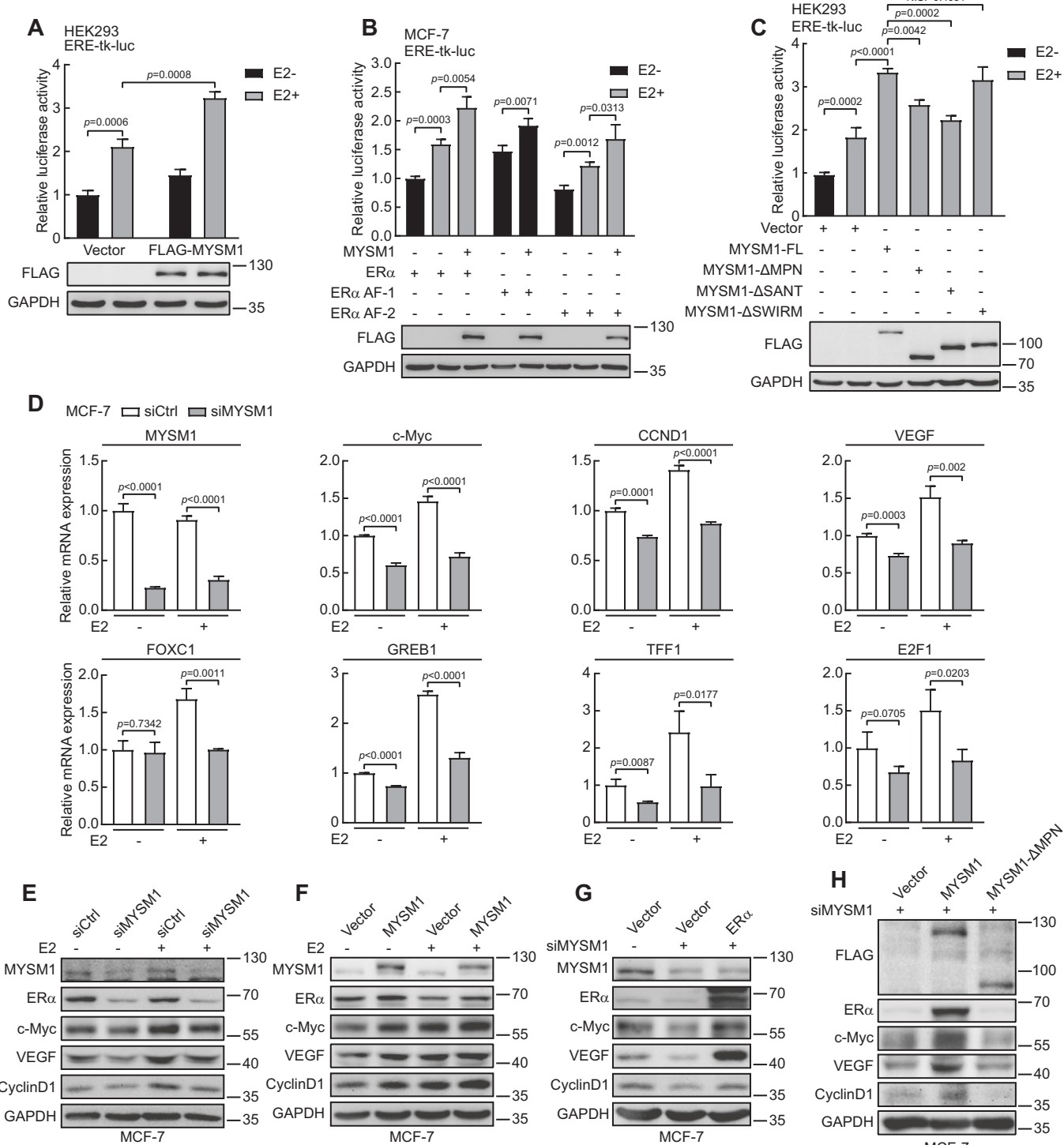

MYSM1-ΔMPN, indicating that MYSM1 is involved in ERα deubiquitination (Fig. 3I). So far, the most thoroughly characterized polyubiquitin processes are lysine 48 (K48)-and K63-linked ubiquitination. K48-linked chains prefer to target proteins for proteasomal degradation, while K63-linked chains act as a molecular glue for complex formation in various signaling pathways and also participate in protein degradation (Grice and

Nathan, 2016; Hayden and Ghosh, 2008; Liu et al, 2018; Madiraju et al, 2022; Wang et al, 2022). We next examined which sites of polyubiquitin chains attached to ERα could be removed by MYSM1. HEK293 cells were transfected with HA-tagged Ubiquitin (Ub) K0, K48, or K63 plasmid for ubiquitination assays. K0 mutant lacking lysine residues was used as a negative control. Ectopic expression of MYSM1 contributes to reduction on K48 and K63

**Figure 2.   MYSM1 enhances ERα-mediated gene transcription in mammalian cells.**

(A) Relative luciferase activities in HEK293 cells transfected with ERα expression plasmid, ERE-tk-luc, pRL-tk, and the indicated expression plasmids in the presence or absence of E2 (100 nM). The expression of MYSM1 was detected by western blot. (B) MYSM1 increases ERα AF1 and ERα AF2 mediated transcriptional activity. MCF-7 cells were transfected with ERα full length or truncated mutants harboring ERα AF-1 or ERα AF-2 with or without MYSM1 expression. (C) The effect of a domain-defective mutation of MYSM1 (ΔMPN, ΔSANT, and ΔSWIRM) on luciferase activity in dual-luciferase reporter system transfected HEK293 cells. The expression of MYSM1 and its truncated mutants were examined with anti-FLAG by western blot. (D) qPCR analysis to determine the transcript amounts of certified ERα target genes in MCF-7 cells with MYSM1-depleted. (E, F) Immunoblot of ERα target gene expression using the indicated antibodies in MYSM1-depleted MCF-7 cells (E) and MYSM1-overexpressed MCF-7 cells (F) with or without E2 (100 nM) treatment for 16-18 h ($n = 3$ independent experiments). (G) The loss of ERα and its target genes in MYSM1-depleted MCF-7 cells can be rescued by ectopic ERα expression. MCF-7 cells were transfected with control siRNA (siCtrl) or siRNA specific against MYSM1 (siMYSM1) followed by PcDNA3.1/ERα expression plasmid. (H) Western blot detecting the protein levels of ERα and its target genes in MYSM1-depleted MCF-7 cells transfected with PcDNA3.1/MYSM1/ MYSM1-ΔMPN expression plasmids. Data information: Results in (A–H) are representatives of three independent experiments performed in duplicate. (A–D): *$P < 0.05$, **$P < 0.01$, ***$P < 0.001$ and N.S. stands for no significance (mean ± SD. Student's $t$ test). Source data are available online for this figure.

polyubiquitin of ERα (Fig. 3J). We then sought to determine the requirement of MYSM1 deubiqitination activity for its ubiquitin linkages cleavage by ubiquitination assay with MYSM1-FL and MYSM1-ΔMPN. Our results demonstrated that catalytically loss of function of MYSM1 mutant (MYSM1-ΔMPN) abrogated the deubiquitination level of K48, K63-linked ubiquitin chains on ERα induced by MYSM1-FL (Fig. 3K). Taken together, our data suggest that MYSM1 removes K48 and K63-polyubiquitin conjugates of ERα via its deubiquitination activity, participating in the maintenance of ERα stability.

Since some deubiquitinases (DUBs) reverse their self-ubiquitination to regulate protein expression of themselves, we wonder whether MYSM1 modulates its own protein stability (Hou et al, 2021). As shown in Appendix Fig. S1A,B, there was no significant change in endogenous or exogenous MYSM1 (MYSM1-HA) expression after transfection with MYSM1-FL or MYSM1-ΔMPN plasmids carrying FLAG tag. These results indicated that MYSM1 might have no effect on its own protein expression.

## MYSM1 is recruited to the cis regulatory element regions of ERα target genes to be involved in histone modification orchestration

To further assess the epigenetic mechanism underlying the modulation of MYSM1 on ERα action, chromatin immunoprecipitation (ChIP) assay was performed to examine the recruitment of MYSM1 or ERα on estrogen response element (ERE) on the upstream of the transcription start site (TSS) of c-Myc, which is a putative ERα target gene (Sun et al, 2020). The results showed that MYSM1 or ERα was recruited to the ERE region of c-Myc upon E2 treatment (Fig. 4A). ChIP assay was further conducted to detect the recruitment of MYSM1 or ERα to estrogen response elements of a number of ERα target genes, including c-Myc, CCND1, VEGF, TFF1 and GREB1 upon MYSM1 knockdown (Sun et al, 2020). We observed that MYSM1 or ERα was recruited to EREs on these genes, and ERα enrichment at ERE regions of these genes displayed a descended trend in MYSM1-depletion cells with E2 treatment (Fig. 4B).

Using the STRING database, we have predicted an interaction network among MYSM1, ERα, and subunits of the classical histone acetyltransferase complex, including pCAF (encoded by *KAT2B*), CBP (encoded by *CREBBP*), and p300 (encoded by *EP300*) (Fig. EV1B). Co-IP experiment results demonstrated the association between ERα and the histone acetyltransferases (HATs) together with MYSM1 in MCF-7 cells. In addition, the results showed that

the association between HATs co-regulatory complex and ERα were diminished after MYSM1 knockdown, suggesting that MYSM1 may potentially associate with the HATs complex to up-regulate ERα action (Fig. 4C).

MYSM1 as a deubiquitinase is involved in the deubiquitination of histone H2A to enhance gene transcription (Fiore et al, 2020). To study the influence of MYSM1 on histone modification on the EREs of ERα target gene, ChIP assays were further performed in BCa cells with siRNA against MYSM1 (siMYSM1). The results demonstrated that MYSM1 or ERα was recruited to the ERE regions on the promoter of c-Myc or CCND1 upon E2 treatment. MYSM1 depletion decreased the recruitment of ERα, pCAF, CBP, and p300 on EREs. Importantly, MYSM1 knockdown enhanced histone H2Aub level, while H2Bub1 level had no significant change. Moreover, the results showed that MYSM1 depletion inhibited H3K4me3, H3K9ac and H3K27ac levels on ERE regions, suggesting that MYSM1 may crosstalk with the HATs complex to modulate histone modifications on ERE regions of ERα-regulatory genes in BCa-derived cells (Fig. 4D; Appendix Fig. S2A–C).

To further determine whether MYSM1 and ERα are recruited together to the promoter of c-Myc, ChIP-re-ChIP assays were performed. The data showed that MYSM1 and ERα are recruited to ERE region of c-Myc promoter at the same time in the presence of E2 (Fig. 4E,F; Appendix Fig. S2D,E). Collectively, the data suggest that knockdown of MYSM1 reduces histone H2Aub level on EREs, and declines ERα recruitment to EREs on ERα target genes.

It's known that the majority fraction of ERα-interaction sites is found at enhancers that are distant from annotated genes, and ERα occupancy on enhancers is a key strategy underlying E2-induced gene expression (Li et al, 2013). Considering the binding of signal-dependent transcription factors on enhancers are generally regulated by related associated cofactors, we turned to determine whether MYSM1 influences ERα occupation on the enhancer regions of traditional ERα target gene (Jiang et al, 2019; Liu et al, 2014). ChIP assays showed a distinct increase of MYSM1, ERα and the HATs recruitment upon c-Myc enhancer in MCF-7 and T47D cells in the presence of estrogen, while MYSM1 knockdown impaired their interactions on enhancer, accompanied by decreased levels of the enhancer hallmarks H3K4me1 and H3K27ac (Fig. EV3A,B). This suggests that MYSM1 acting as a functional cofactor affects the binding of ERα and its co-regulatory complex to ER-related enhancer. ERα causes a global increase in enhancer RNA (eRNA) transcription on enhancers adjacent to E2-upregulated coding genes. As functional transcripts, these eRNAs

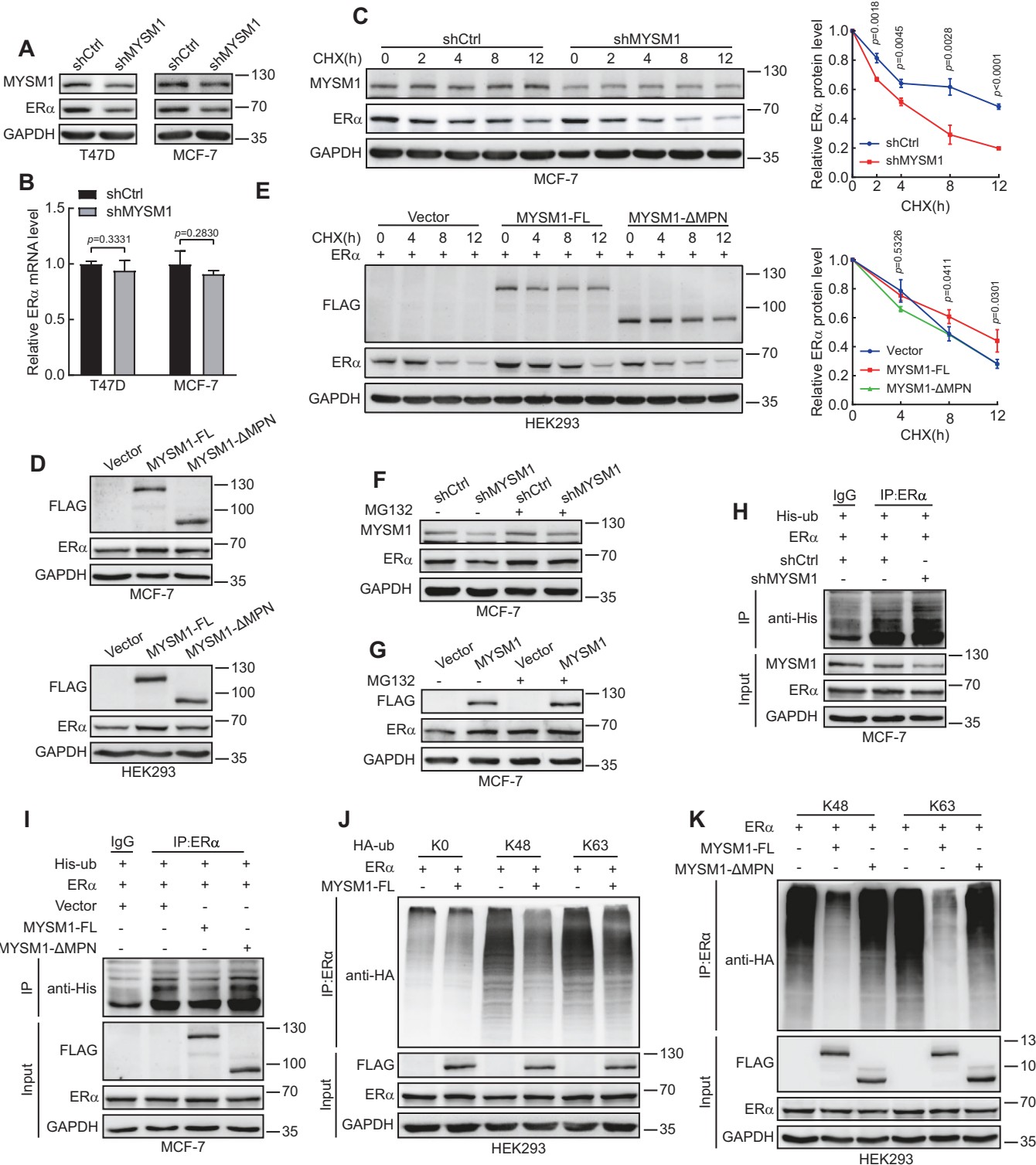

are predictive marks of enhancer activity and induce the observed ligand-dependent target-coding genes that in turn define cell fate (Wang et al, 2011). qPCR was performed to examine how the recruitment of MYSM1 controls transcriptional activity of ERα-binding enhancers. MYSM1 depletion significantly reduced the eRNA expression of *c-Myc*, *CCND1*, *E2F1*, *GREB1*, and *TFF1*, indicating that MYSM1-recruited enhancers exhibited increased

transcriptional activity (Fig. EV3C). It is well-recognized that apart from pioneer factors, the localization of co-activator complex at promoter and enhancer regions could alleviate the chromatin accessibility limitation of transcription factors and RNA polymerase on DNA (Jiang et al, 2019). Strengthening ERα-chromatin interactions could govern the genomic circuitry by inducing massive transcriptional activation. As MYSM1 protein contains

**Figure 3. MYSM1 stabilizes the ERα protein through ERα deubiquitination.**

(A, B) Western blot (A) and qPCR (B) analysis in T47D/MCF-7 cells infected with shCtrl or shMYSM1 lentivirus to evaluate the impact of MYSM1 depletion on ERα protein and mRNA levels. (C) MCF-7 cells carrying control shRNA or shMYSM1 were complemented with CHX (20 μM) for particular time periods, and cell lysates were assessed by western blot. Relative ERα level was quantified by densitometry and presented in the right plot. (D) Western blot analysis of ERα and exogenous MYSM1 expression in MCF-7/HEK293 cells transfected with wild-type MYSM1 plasmid or MPN-deletion mutant. (E) Cell lysates from MYSM1-FL/MYSM1-ΔMPN overexpressing MCF-7 cells stimulated with CHX (20 μM) at specified time points were denatured and quantitated by western blot. Relative ERα level was quantified by densitometry and presented in the right plot. (F, G) Extracts obtained from MCF-7 cells harboring shMYSM1 or FLAG-MYSM1 upon MG132 (10 μM) treatment for 4 h were analyzed by western blot. (H) MCF-7 cells from the control or the MYSM1-knockdown group were harvested after MG132 (10 μM) addition for 6 h, followed by immunoprecipitation with anti-ERα and subsequently probed with anti-His. (I) Immunoblot analysis of the polyubiquitination of ERα proteins in MCF-7 cells transiently co-transfected with plasmids encoding ERα and FLAG-MYSM1 or MYSM1-ΔMPN. MG132 (10 μM) was added to cell culture 6 h before cell collection. Cell lysates were subjected to immunoprecipitation with anti-ERα and immunoblot with anti-His. (J, K) HEK293 cells transfected with the indicated plasmids were extracted and immunoprecipitated with anti-ERα, followed by immunoblot with anti-HA. Data information: (A–K): $n = 3$ independent experiments performed in duplicate. (A, B, D): mean ± SD; Student's *t* test; $n = 3$, NS: not significant. Source data are available online for this figure.

the SANT domain, which is broadly represented among chromatin-remodeling enzymes, we speculated that MYSM1 may implicated in chromatin decondensation to exert its co-activator function (Boyer et al, 2004). Micrococcal nuclease (MNase) experiments demonstrated that MYSM1 knockdown in MCF-7 cells weakened MNase accessibility and the extent of chromatin relaxation is reduced, while MYSM1 overexpression increased the quantity of cleaved chromatin, suggesting that MYSM1 probably participates in maintaining chromatin accessibility (Fig. EV3D,E).

## MYSM1 depletion and its docking molecule Imatinib inhibits cell growth through MYSM1-ERα axis in ERα-positive BCa-derived cells

Having established the molecular mechanisms of MYSM1 on ERα action, we then turned to interrogate the potential biological function of MYSM1 on ER-positive breast cancer progression. The results from colony formation assay showed that MYSM1 depletion suppressed colony formation in MCF-7 and T47D cells (Figs. 5A and EV4A). Consistently, growth curve analysis showed that knockdown of MYSM1 inhibited cell proliferation, and ectopic expression of ERα partly rescued this condition (Figs. 5B and EV4B). We then performed flow cytometry for detecting the influence of MYSM1 on cell cycle as displayed. The results showed that MYSM1 depletion retarded the G1-S phase transition (Figs. 5C and EV4C–E). Above all, the results indicate that MYSM1 promotes cell proliferation and G1-S phase transition in BCa, and the effect of MYSM1 on cell growth is partially related to ERα pathway.

To test the functional of MYSM1 on BCa cell growth in vivo, MCF-7 cells infected with shMYSM1 or negative control lentivirus (shCtrl) were individually implanted subcutaneously into flank sides of 4-week-old female *BALB/c* nude mice ($n = 7$) armpit. We monitored tumor growth and measured tumor sizes every 5 days after injection. Compared with shCtrl-MCF-7 cells, shMYSM1-MCF-7 cells carried decreased tumor growth rates (Fig. 5D). In accordance with the growth curve, tumors originating from shMYSM1-MCF-7 cells exhibited lower weights and smaller sizes than those from the control (Fig. 5E–G). Moreover, immunohistochemical (IHC) analysis was performed to examine the expression in ERα and its target expression. The results revealed that loss of MYSM1 dramatically decreased ERα and c-Myc protein expression, accompanied by a reduction in the Ki67 index in xenograft tumor tissue (Figs. 5H and EV4F). Taken together, these results demonstrated that MYSM1 depletion suppressed the breast cancer cell growth in mice.

Given its pleiotropic biological functions, MYSM1 may be a potential therapeutic target for drug development against breast cancer, especially in endocrine resistant ERα-positive breast cancer. The combination of drug repurposing and virtual screening of structure-based compound libraries has greatly facilitated the development of anticancer drugs. We thus focused on the virtual screening of the commercially-available compounds retrieved from the ZINC database (https://zinc.docking.org/) to find the compounds that could spatially interact with the MPN domain of MYSM1 protein (AlphaFold ID: AF-Q5VVJ2-F1) (Irwin et al, 2020; Jumper et al, 2021). The compounds library was subjected to virtual screening using the molecular operating environment (MOE).

The screened molecules were prioritized according to the score of combined energy, the top 15 molecules were selected for list (Table 1). According to their characteristics and the molecule-protein binding affinity, four molecules (Deferoxamine mesylate, Nilotinib, Imatinib, and Candesartan Cilexetil) were picked up for cell cytotoxicity tests. The results showed that Imatinib and Nilotinib exhibited the most obvious growth inhibition as the concentration reached to 1 μM (Fig. EV4G). Then we performed luciferase assay to compare the effects of these two compounds on MYSM activity. The results indicated that both drugs inhibited the up-regulation of ERα action by MYSM1, while Imatinib was more effective, thus we continued utilizing Imatinib as a MYSM1 inhibitor for further investigation (Fig. EV4H). As shown in Fig. 5I, Imatinib diminished the activity of MYSM1 as an ERα co-activator in a dose-dependent manner in MCF-7 cells. To explore the segment of Imatinib acting on MYSM1, we utilized PyMOL molecular graphics system to dock this small molecule candidate into the corresponding binding site of MYSM1. As expected, molecular docking results predicted that Imatinib is mainly linked with MYSM1 at amino acids GLU-597 and ARG-534 in the form of hydrogen bonds, of which GLU-597 is located in the MPN domain (Fig. 5J). Luciferase assay was performed to further confirm the effect of Imatinib on MPN domain activity. In the control group, the loss of MPN domain contributed to a decreased transcriptional activity (lane1 vs. lane2). In the Imatinib-treated group, there was no significant difference in ERα-induced gene transcription after transfection with MYSM1-FL or MYSM1-ΔMPN (lane3 vs. lane4), demonstrating that Imatinib could regulate MYSM1 activity through the MPN domain (Fig. EV4I). Western blot experiments also showed that Imatinib addition reduced the expression of ERα and its downstream target genes (Fig. EV4J). Finally, we performed MTS and colony formation assays to investigate the effect of Imatinib in combination with shMYSM1 on cell viability in BCa

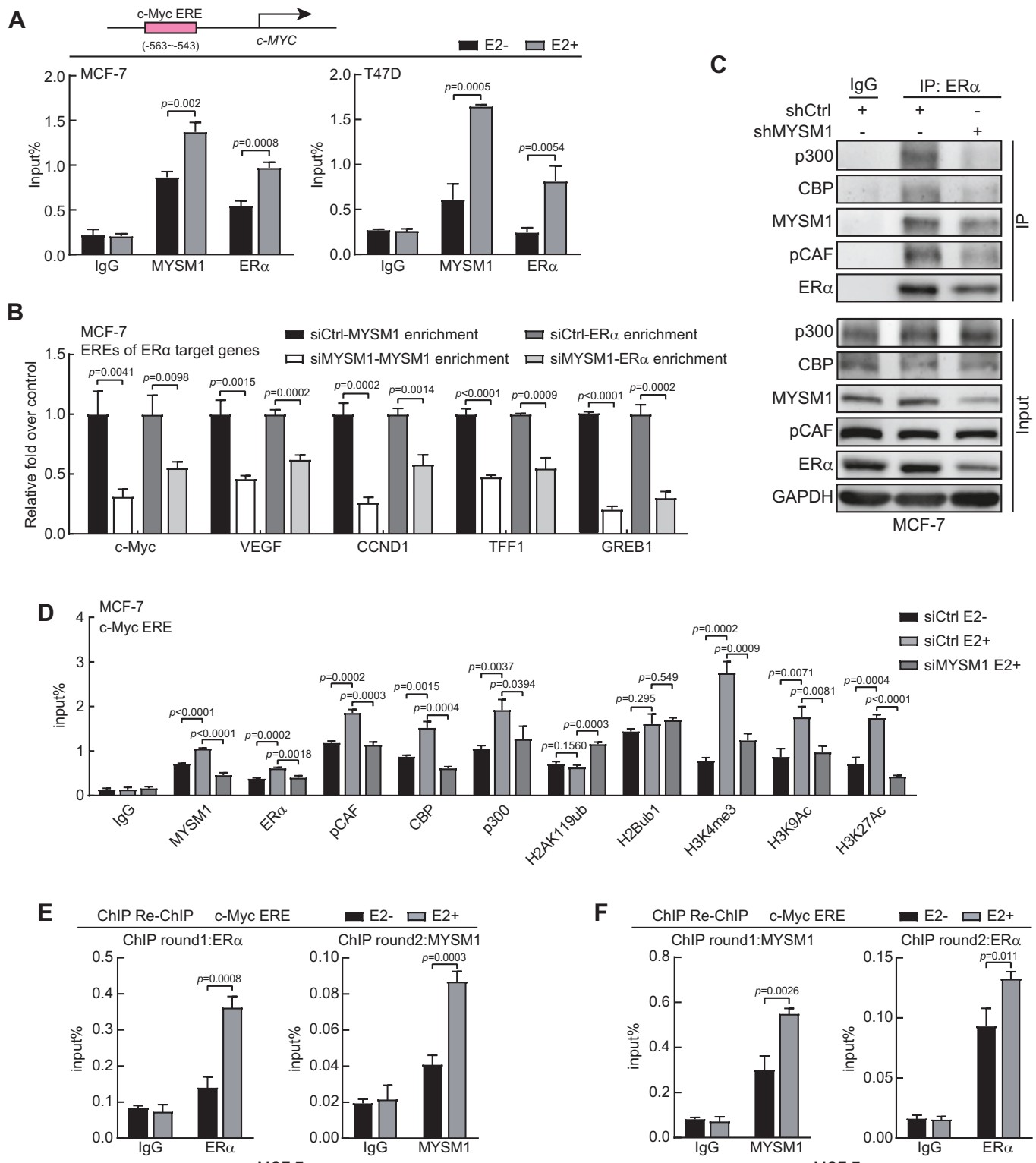

cells. Both shMYSM1 and Imatinib alone restrained cell growth, and their co-application had a more significant effect (Figs. 5K and EV4K–M). Above all, these results implied that Imatinib could exert its inhibitory action on breast cancer cell proliferation through the MYSM1-ERα axis.

## MYSM1 depletion facilitates the sensitivity of ERα-positive breast cancer cells to antiestrogen treatment

Estrogen inhibitors or ERα antagonists dependent on the estrogen-ERα axis have ruled supremely for decades to treat ERα-positive

**Figure 4.  MYSM1 facilitates the recruitment of ERα and HAT complex at the cis elements of ERα target genes in MCF-7 cells.**

(A) The top half is the schematic diagram of the putative estrogen response element (ERE) of c-Myc. The bottom half is ChIP assays with indicated antibodies showing the recruitment of MYSM1 and ERα at c-Myc ERE. (B) MYSM1 and ERα ChIP analysis was conducted in MYSM1-deficiency MCF-7 cells on some ERα-binding sites of E2-induced genes as indicated. Cells were cultured in estrogen-depleted medium for 48 h before stimulation with 100 nM E2 for 2 h. The immunoprecipitated DNA fragments were analyzed by qPCR using primers recognizing the promoter region of ERα target genes. The amplified products were standardized by a certain amount of unprecipitated input DNA. (C) Co-immunoprecipitation was performed with anti-ERα or IgG in MCF-7 cells carrying control shRNA or shMYSM1. Precipitated proteins were determined by western blot using antibodies against acetyltransferase core proteins p300/CBP/pCAF or MYSM1/ERα as indicated. (D) ChIP assays via designated antibodies to elaborate the influence of MYSM1 regarding to ERα recruitment and relative histone modifications on *c-Myc* ERE in MCF-7 cells. (E, F) ChIP re-ChIP experiments validated that E2 treatment augmented concurrent recruitment of MYSM1 and ERα on *c-Myc*-ERE in MCF-7 cells. Chromatin extracts immunoprecipitated with anti-ERα (E) or anti-MYSM1 (F) as the round1 antibody were re-immunoprecipitated with anti-MYSM1 (E) or anti-ERα (F), respectively, followed by qPCR to calculate the collected DNA signals. Data information: (A–F); n = 3 independent experiments performed in duplicate. (A, B, D–F): *P < 0.05, **P < 0.01, ***P < 0.001, N.S. means no significance (mean ± SD; Student's *t* test; n = 3). Source data are available online for this figure.

breast cancer. Letrozole belonging to aromatase inhibitors (AIs), Tamoxifen acting as a selective ERα modulator (SERM), and Fulvestrant as a selective ERα degrader (SERD) are used as the putative ERα antagonists. Having established that MYSM1 participates in modulation of ERα signaling pathway, we postulate that MYSM1 might be implicated in endocrine resistance process. We thus turned to collect the clinical ERα-positive breast cancer patients with short-term AI adjuvant treatment (12 weeks). The pathological features of all the samples were non-gestational ER+/Her2− invasive breast cancer at stage I to III. Premenopausal patients were treated with Ovarian function suppressor (OFS) plus AI and postmenopausal patients were treated with AI monotherapy. Western blot was performed to detect MYSM1 expression in biopsy samples of these patients before and after AI adjuvant treatment. According to Ki67 index and breast MRI (Magnetic Resonance Imaging) clinical features, these patients were separated into responders (Ki67 ≤ 10%) and non-responders (Ki67 > 10%). The grayscale analysis in two cohorts as indicated showed that MYSM1 protein was up-regulated in the non-responders after AI adjuvant treatment, suggesting that MYSM1 may act as a potential predictor of clinical AI therapy sensitivity. (Figs. 6A and EV5A). Proliferation marker Ki67 is the prototypic cell cycle related nuclear protein (Scholzen and Gerdes, 2000). Ki67 expression is a criterion for determining the efficacy of neoadjuvant treatment in AI resistance patients, and the variability in Ki67 levels could be correlated with cell cycle regulation. We thus set out to explore the potential link between MYSM1 expression and cell cycle progression (Sobecki et al, 2017). MCF-7 cells were subjected to serum deprivation for 2 days to induce cell cycle synchronization. After that, the arrested cells were released into cell cycle by serum replenishment for different times. Western blot experiments showed that CCND1, the cell cycle regulator, gradually accumulated as the duration of serum re-feeding extended, and the expression of MYSM1 had a similar trend (Fig. 6B). Moreover, the CCND1 expression levels were also consistent with those of MYSM1 in the AI adjuvant-treated samples, indicating that MYSM1 protein expression might be related to the cell cycle state (Figs. 6C and EV5A).

To examine the influence of MYSM1 expression on the antiestrogen effects, cell viability was evaluated in MYSM1-silencing MCF-7 cells with ERα antagonist treatment. Compared with the group of E2 existence only, MYSM1 depletion exacerbates cell death with an extra addition of appropriate concentration of Tamoxifen, Fulvestrant, or Letrozole. These results suggest that MYSM1 may confer the resistance to antiestrogen treatment (Figs. 6D,E and EV5B–E). Besides, cell

growth curves and colony formation proved that MYSM1 depletion made cancer cells more susceptible to the drug toxicity effects under the condition of higher dosage of specific endocrine drugs (Figs. 6F,G and EV5C,D,F,G). BT474 cells owing intrinsic Tamoxifen resistance were also used to construct cell lines stably infected with shMYSM1 (Fig. 6H). Figure EV5H,I showed that MYSM1 deficiency facilitated the sensitivity to Tamoxifen treatment in BT474 cells. Furthermore, we introduced the Tamoxifen-resistant MCF-7 (MCF-7 TMR) and T47D (T47D TMR) cell lines to confirm the effect of MYSM1 on cellular antiestrogen sensitivity. MYSM1 protein levels were significantly higher in MCF-7 TMR and T47D TMR cells than those in the corresponding parent cells, which is in accordance with its expression in the AI adjuvant-receiving BCa patients (Fig. EV5J). To explore the biological effects of MYSM1 on Tamoxifen resistance, we established MYSM1-knockdown MCF-7 TMR and T47D TMR cell lines to carry out MTS and colony formation assays (Fig. EV5J). Compared with the control group, MYSM1-depleted TMR cells exhibited restored Tamoxifen sensitivity (Figs. 6I,J and EV5K,L). Finally, we examined whether Imatinib, the predicted MYSM1-binding compound, is involved in the reverse of Tamoxifen resistance. The growth curve showed that the combination of Imatinib with Tamoxifen achieved a more markable inhibition of MCF-7 TMR cells growth than Tamoxifen or Imatinib alone (Fig. 6K). As expected, MCF-7 TMR cells grown without Imatinib, were resistant to Tamoxifen treatment. However, after supplementation with 1 μM Imatinib, which resulted in about 15% growth inhibition, Tamoxifen treatment achieved an impressive blocking effect on cell proliferation dose-dependently, demonstrating that treatment with Imatinib renders the resistant cells sensitive to Tamoxifen inhibition (Fig. 6L).

## Expression of MYSM1 is upregulated in clinical breast cancer tissues

The regulation function of MYSM1 on ERα-induced transactivation indicates that MYSM1 acts as a novel ERα co-activator, suggesting that MYSM1 may play an important role in breast cancer. We then conducted western blot and IHC experiments to estimate MYSM1 expression and the correlation between MYSM1 expression and clinicopathologic factors of the patients. Western blot was performed with specimens from 30 pairs of fresh ERα-positive breast cancer tissues and the adjacent benign mammary tissues. MYSM1 protein is prominently up-regulated in breast cancer cohort (Fig. 7A,B). Moreover, its expression was positively correlated with that of ERα (Fig. 7A,C). qPCR results in 9 pairs of breast cancer individuals also showed an internal

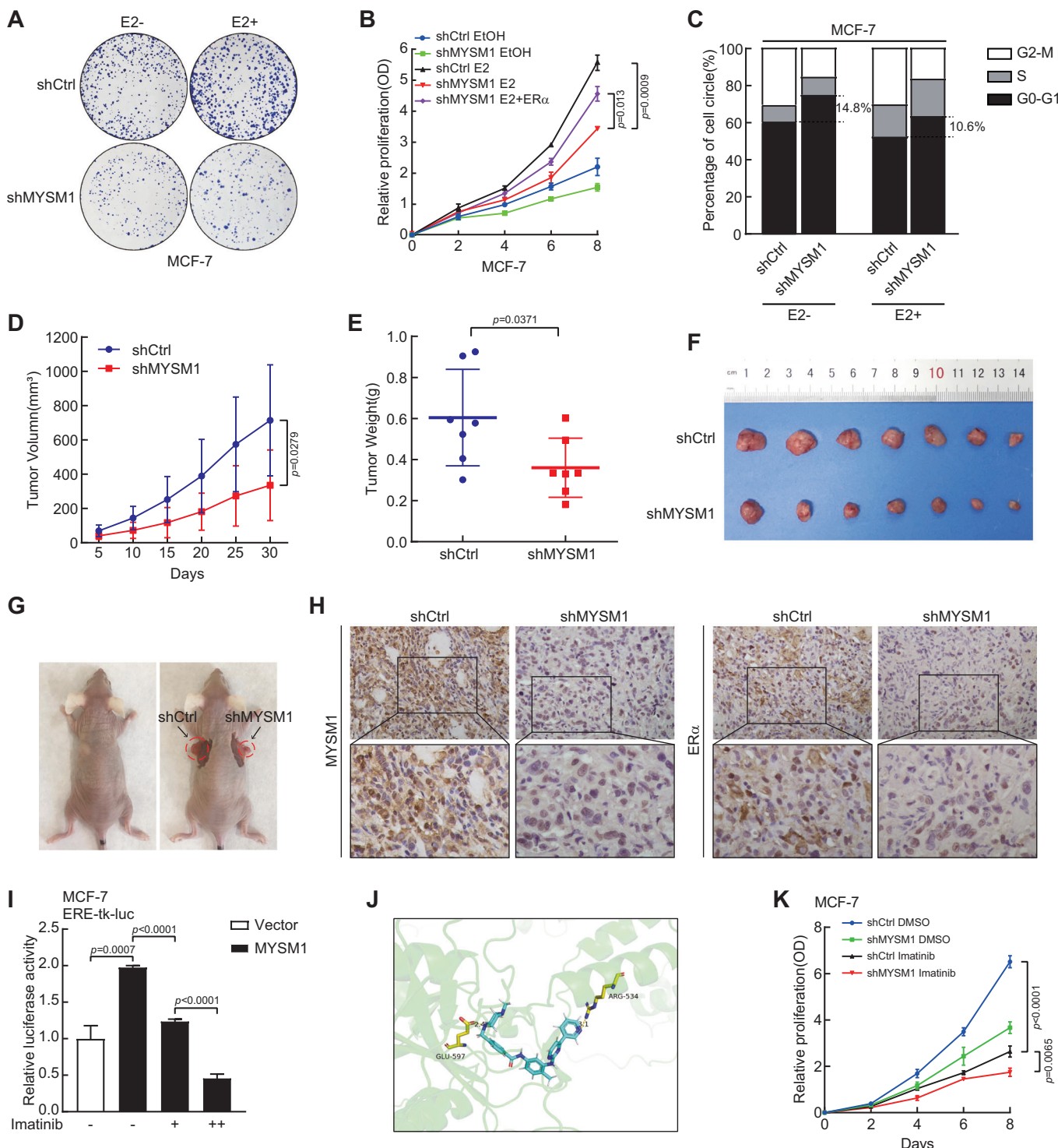

consistency of the transcriptional level of MYSM1 as its mRNA levels are higher in the tumor group (Fig. 7D). To sketch out the clinical relevance of MYSM1 in breast cancer, we took advantage of the commercial tissue microarrays as well as some paraffin sections owing detailed patient information sponsored by hospital. The results from IHC staining with the anti-MYSM1 antibody demonstrated that the expression of MYSM1 gradually increased as the BCa proceeded into higher grade (Fig. 7E,F). The statistical

analysis of clinicopathological characteristics proposed a tight bond in terms of MYSM1 expression with histological grade, ERα status, and HER2 status (Table 2). Finally, patients were split into two groups based on the median MYSM1 expression value to calculate the survival rates. The data showed that higher expression of MYSM1 exhibited the poor overall survival, implying that MYSM1 may act as an indicator in breast cancer prognosis (Fig. 7G).

◀ **Figure 5.   MYSM1 depletion or the small molecule Imatinib docked to MYSM1 suppresses breast cancer cell growth through MYSM1-ERα axis.**

(A) Colony formation assay of shCtrl and shMYSM1 stably expressed MCF-7 cells cultured under vehicle or E2 (100 nM) treatment for 15 days. (B) Growth curve showing the effect of MYSM1 knockdown on MCF-7 cell proliferation with or without E2 (100 nM). Total cell viability was assessed every other day by MTS assay. (C) Representative histogram displaying percentage of the cell population in G0-G1, S, and G2-M phases in MCF-7 cells carrying shCtrl or shMYSM1 under E2 stimulation or not, as detected by flow cytometry. (D) Tumor xenografts were generated by injecting MCF-7 cells infected with shCtrl and shMYSM1 subcutaneously in female *BALB/c* mice. The average tumor volume was measured at the indicated time point. (E) Tumor weights of the shCtrl and shMYSM1-group were measured 30 days after cell injection. (F, G) Representative images of the dissected tumors harboring shCtrl and shMYSM1-expressed MCF-7 cells. (H) Paraffin sections of the nude mice tumors were conducted to immunohistochemistry (IHC) staining with anti-MYSM1 and anti-ERα. The representative pictures were taken at ×20 magnification of microscopic field. (I) Relative luciferase activities in MCF-7 cells transfected with ERα expression plasmid, ERE-tk-luc, pRL-tk, along with PcDNA3.1 or MYSM1-FL plasmids in the presence of E2 (100 nM) with the gradually increased concentration of Imatinib (2 μM or 4 μM). (J) Localized views of MYSM1-Imatinib composites obtained by molecular docking. (K) Growth curve showing the effect of shMYSM1, Imatinib, or their combination (1μM) on MCF-7 cell proliferation. Total cell viability was assessed every other day by MTS assay. Data information: **$P < 0.01$, ***$P < 0.001$ (mean ± SD; Student's $t$ test; (A–C, F–K): $n = 3$ independent experiments; (D, E): $n = 7$). Source data are available online for this figure.

**Table 1.   Small-molecule protein-binding affinity evaluation based on MOE (kcal/mol).**

| ZINC ID | Ligand name | Docking score |
|---|---|---|
| ZINC000003830276 | Benzonatate | −10.7336 |
| ZINC000070466416 | Cabozantinib | −10.1654 |
| ZINC000095564694 | Naloxegol | −10.1551 |
| ZINC000003943279 | Dabigatran-etexilate | −10.0238 |
| ZINC000003830635 | Deferoxamine mesylate | −9.987 |
| ZINC000006716957 | Nilotinib | −9.8035 |
| ZINC000035328014 | Ibrutinib | −9.8034 |
| ZINC000004095858 | α-Vitamin E | −9.7675 |
| ZINC000019632618 | Imatinib | −9.7329 |
| ZINC000004102194 | Mupirocin | −9.7312 |
| ZINC000003989268 | Ceftaroline fosamil | −9.7301 |
| ZINC000095619101 | Ranolazine | −9.7183 |
| ZINC000003871832 | (S)-Verapamil hydrochloride | −9.6485 |
| ZINC000003812888 | (R)-Verapamil hydrochloride | −9.625 |
| ZINC000004097427 | Candesartan Cilexetil | −9.4949 |

## Discussion

The ERα signaling acts as an obligate component in female mammary growth and development. It requires strict supervision to prevent against blunted mammary glands morphogenesis or the tumorigenic processes (Rusidze et al, 2021). The extraordinarily higher expression of ERα protein in breast cancer tissues reflects a specific genomic environment and intricate signaling networks in tumor cells. In this study, we identified a histone deubiquitinase, MYSM1 as a novel ERα co-activator in breast cancer. Our results have demonstrated that MYSM1 is recruited together with ERα to the promoter region of ERα target genes, leading to epigenetic modulation of H2Aub, H3K4me3, H3K9ac, and H3K27ac levels. On the other hand, we provided the evidence that MYSM1 is involved in maintenance of ERα stability via reduction of Lysine 48 (K48) and K63-linked poly-ubiquitination on ERα. Thus, MYSM1 enhances ERα action via histone and non-histone deubiquitination to promote cell proliferation and antiestrogen insensitivity in breast cancer progression. Our study suggests that MYSM1 as a deubiquitinase is involved in up-regulation of ERα action, exerting epigenetic modifier and ERα stability maintainer to enhance antiestrogen insensitivity in ERα positive breast cancer.

Modification of histone tails by epigenetic regulators results in reprogramming of chromatin landscape to regulate gene transcription (Zhao et al, 2021). Histone H2A ubiquitination is a highly conserved histone modification that compacts chromatin structure and decreases chromatin accessibility, thereby reducing the specific affinity of transcription factors to inhibit transcriptional regulation of non-lineage specific target genes (Barbour et al, 2020; Higashi et al, 2010; Tamburri et al, 2020). Uncontrolled H2Aub deposition leads to genomic instability and is prevalent in cancer development (Tamburri et al, 2021). MYSM1 as a histone H2A deubiquitinase has been reported to activate the transcription of its downstream genes, including CDH1, c-MET, or genes encoding ribosomal proteins via histone H2A deubiquitination in colorectal cancer, melanoma, and B cell lymphoma (Chen et al, 2021; Lin et al, 2021; Wilms et al, 2017). Consistent with these studies, our results indicate that MYSM1 enhances ERα-induced transactivation by eliminating H2Aub deposition. In addition, MYSM1 appears to motivate the accumulation of active epigenetic marks H3K4me3, H3K9ac and H3K27ac. In fact, previous studies have demonstrated that MYSM1 interplays with p/CAF to form a co-regulatory protein complex that stepwise synergizes histone ubiquitination and acetylation for AR-dependent gene activation in prostate cancer cells (Zhu et al, 2007). In this study, we provided the evidence to show that MYSM1 is required for the association between the HATs complex and ERα, modulating active histone modifications to enhance chromatin accessibility in a serial and combinatorial manner in the breast cancer cellular context (Figs. 4C,D and EV1B).

By cleaving ubiquitin chains from non-histone substrates, deubiquitinase could accelerate tumor development by modulating the expression, activity, and localization of many substrates (Pal et al, 2014). ERα has been shown to be a substrate of deubiquitinase (Pesiri et al, 2016). A few DUBs exert their pro-oncogenic function by triggering ERα deubiquitination to maintain the stability of ERα in BCa (Cao et al, 2021; Tang et al, 2021; Wang et al, 2020; Xia et al, 2021; 2019). Furthermore, DUB inhibitors targeting USP14 and UCHL5 downregulate ERα expression and induce cancer cell apoptosis, providing a prospective strategy for ER-positive BCa treatment (Xia et al, 2018). Due to the characteristic of a higher turnover rate and greater homeostasis demand of ERα protein in breast cancer cells, ERα-targeting DUBs have become critical targets for novel cancer therapeutics discovery (Xiao et al, 2016). As

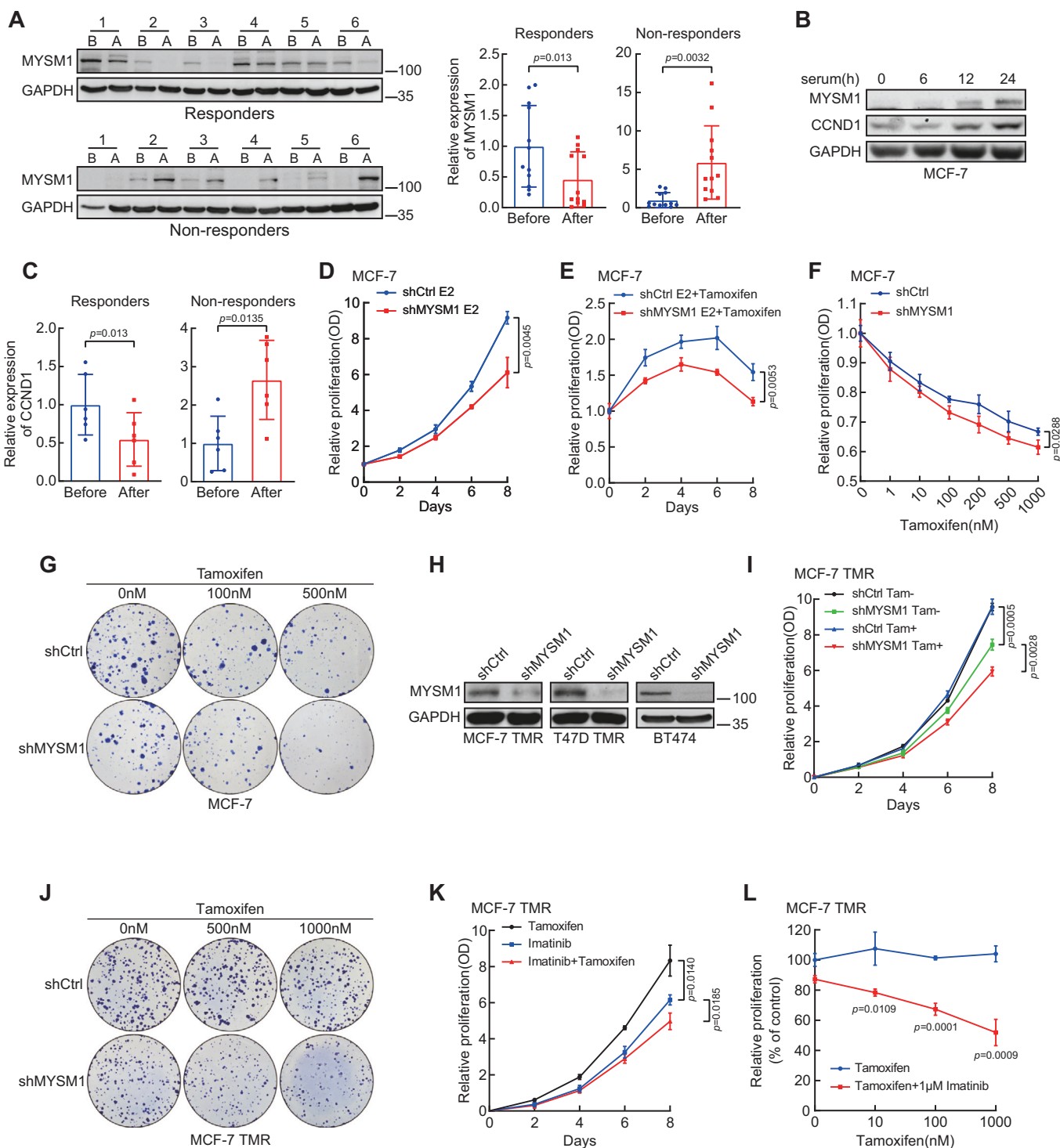

a ubiquitin-specific protease, MYSM1 was reported to remove K63-linked polyubiquitin chains of TRAF3, TRAF6, RIP2, and STING, resulting in the collapse of the complex scaffolds and attenuated complex assembly, thereby attenuating the associated signaling pathways (Panda and Gekara, 2018; Panda et al, 2015; Tian et al, 2020b). However, its catalytic activity towards polyubiquitin chains of different geometric configurations on different substrates remains unclarified. Herein, our data demonstrated that MYSM1

removes the K48- and K63-linked polyubiquitination on ERα and maintains its protein stability (Fig. 3J,K). K48-linked polyubiquitination induces degradation of proteasomal substrates, and K63-linked polyubiquitination mainly modulates non-proteolytic cellular processes, such as signal transduction. One study found that K63 chains conjugated to TXNIP could attract K48 ligases to trigger K48/K63 branched chains formation and regulates proteasomal degradation, providing a new insight for heterogenous ubiquitin

**Figure 6. MYSM1 depletion enhances the sensitivity of ERα-positive breast cancer cells to antiestrogen treatment.**

(A) ERα-positive BCa patients that received AI adjuvant treatment were divided into 2 groups named "Responders" and "Non-responders". MYSM1 protein expression in clinical biopsy samples before and after treatment were examined by western blot. "B" represents cases before AI treatment, "A" represents cases after AI treatment. Semiquantitative analyses are presented as relative expression of MYSM1 normalized to that of the before-group. (B) Western blot showing course of MYSM1 and CCND1 expression after 48 h serum starvation (0 h), and at serial intervals following serum replenishment (6, 12, 24 h) in MCF-7 cells. (C) Boxplots displaying relative expression of CCND1 in the "After" group normalized to that of the "Before" group from the AI adjuvant treatment-receiving patients. (D, E) Growth curve showing the effect of MYSM1 knockdown on MCF-7 cell proliferation upon Tamoxifen (500 nM) absence (D) or presence (E). Total cell viability was assessed every other day by MTS assay. (F) A cellular viability detection in MYSM1-deletion MCF-7 cells that incubated in various concentrations of Tamoxifen for 7 days. (G, J) The panels show colony-formation assay conducted in MCF-7 (G) or MCF-7 TMR (J). Cells infected with lentivirus expressing shCtrl/shMYSM1. Cells in each panel were treated with different doses of Tamoxifen for 15 days before fixation and R250 staining. (H) Western blot experiment to validate MYSM1-knockdown efficiency in MCF-7 TMR, T47D TMR, and BT474 cells. (I) Growth curve showing the effect of MYSM1 knockdown on MCF-7 TMR cell proliferation with or without Tamoxifen (1 μM). Total cell viability was assessed every other day by MTS assay. One week before seeding, Tamoxifen was withdrawn from the medium of the resistant cell lines. (K) Growth curve showing the cell viability of MCF-7 TMR treated with Tamoxifen (1 μM), Imatinib (1 μM), or their co-application. (L) Cell viability assay in MCF-7 TMR cells treated with the indicated doses of Tamoxifen (0, 10, 100, 1000 nM) in the absence or presence of Imatinib (1 μM). One week before seeding, Tamoxifen was withdrawn from the medium of the resistant cell lines. The results are expressed relative to the control (0 nM Tamoxifen treated). Data information: *$P < 0.05$, **$P < 0.01$, ***$P < 0.001$ (mean ± SD; Student's $t$ test; (A); $n = 12$ independent experiments, (C): $n = 6$ independent experiments, (B, D–L): $n = 3$ independent experiments). Source data are available online for this figure.

chains to convert a non-degradative ubiquitin into a degradation signal (Ohtake et al, 2018). So far, it's unclear whether branched chains could be assembled on ERα protein or the exact function of K63-linked polyubiquitination on signal transduction of ERα pathway. The branched chains represent an increasing level of attached ubiquitin concentrating closely to the substrate for a higher affinity of effectors towards ubiquitin. That could explain the extremely powerful proteolytic signal of these conjugates. The multiple blocks at both types of K48- and K63- linkages by MYSM1 would make its pro-stabilizing effect on ERα more intense based on the assumption of ERα degradation through K48/K63 branched chains.

Aberrant activation of ERα signaling is a common incentive of endocrine resistance in BCa, which may be induced by peculiar post-translational modification of ERα protein, abnormal expression of ERα co-regulators, compensatory cross-talk between ERα and parallel oncogenic signaling pathways, or mutations in ESR1 gene (Belachew and Sewasew, 2021; Hanker et al, 2020). These aspects result in the unconventional expression of ERα signature genes involved in diverse biological processes to oppose endocrine therapy (Louie et al, 2010; Miller et al, 2011). Herein, we provided the evidence to show that MYSM1 co-activates ERα action via histone and non-histone manner to confer antiestrogen insensitivity in breast cancer. MYSM1 up-regulated ERα downstream genes, such as c-Myc, VEGF, and CCND1, which play crucial roles in endocrine resistance, suggesting that MYSM1 may act as an ERα co-activator to be involved in conferring endocrine resistance in BCa (Fig. 2). Moreover, ChIP analysis showed that MYSM1 is recruited to the ERE elements of ERα-regulated genes, which exerts extensive biological functions in pathophysiological processes (Fig. 4). On the other hand, our results also demonstrated that MYSM1 accelerates cell proliferation even in the absence of E2, suggesting that MYSM1 may also participate in regulating non-canonic ERα signaling pathway independent on E2 treatment to participate in promoting breast cancer process (Fig. 5A,B). Moreover, we also provided the evidence to show that MYSM1 was highly expressed in non-responders after AI adjuvant treatment in ERα-positive BCa samples (Fig. 6A). The influence of MYSM1 on the effect of antiestrogen treatment demonstrated that MYSM1 depletion or identified MYSM1 inhibitor (Imatinib) increased the sensitivity of antiestrogen resistant BCa cells to the treatment of antiestrogen drugs (Figs. 5 and EV4G–M). Thus, in this study, we identified MYSM1 as a histone deubiquitinase is a novel ERα co-

activator involved in regulation of histone modifications and ERα deubiquitination, thereby up-regulating ERα-mediated transactivation and maintaining ERα itself stability to participate in enhancement of antiestrogen insensitivity in ERα-positive BCa.

Collectively, our data have demonstrated the function of MYSM1 on maintenance of ERα stability and modulation of ERα action to confer antiestrogen insensitivity in ERα-positive breast cancer, providing a potential therapeutic target for endocrine resistance in ERα-positive BCa.

## Methods

### Drosophila strains and genetics

Fly stocks were maintained on cornmeal sucrose-based media at 25 °C. The human ERα and MYSM1 cDNA cloning products were introduced in a modified pCaSpeR3 vector with upstream active sequence (UAS) promoter to generate the UAS-linked ERα/MYSM1 expression construct. For the reporter construct, ERE sequence copies were integrated upstream of the GFP reporter genes controlled by the TATA promoter box. The UAS-ERα/MYSM1 expression and ERE-GFP reporter constructs were sent to EMBL Drosophila Injection Service for transgenic flies generation. The CG4751 deletion mutant (CG4751$^{-/+}$) were obtained from Bloomington Drosophila Stock Center. To examine the effect of CG4751 loss or MYSM1 gain of function on ERα-dependent transactivation of reporter gene, we crossed the male F0 owing hemizygous mutants (deletion or overexpression) with Gal4-UAS ERE-GFP female flies. F1 progeny possessing the mutant allele and mosaic red eye were carried out for eye disc histology analysis as previously described (Sun et al, 2016).

### Plasmids

Full-length cDNA for MYSM1 was amplified by PCR using the cDNA library as a template. The coding sequences of various MYSM1 mutants amplified using the primers listed in Table EV1. The PCR products for MYSM1 wild-type and a series of mutants were cloned into a p3×FLAG-CMV-10 expression vector to generate MYSM1-FL, MYSM1-ΔMPN, MYSM1-ΔSANT and MYSM1-ΔSWIRM. Other plasmids were identical to our previous studies.

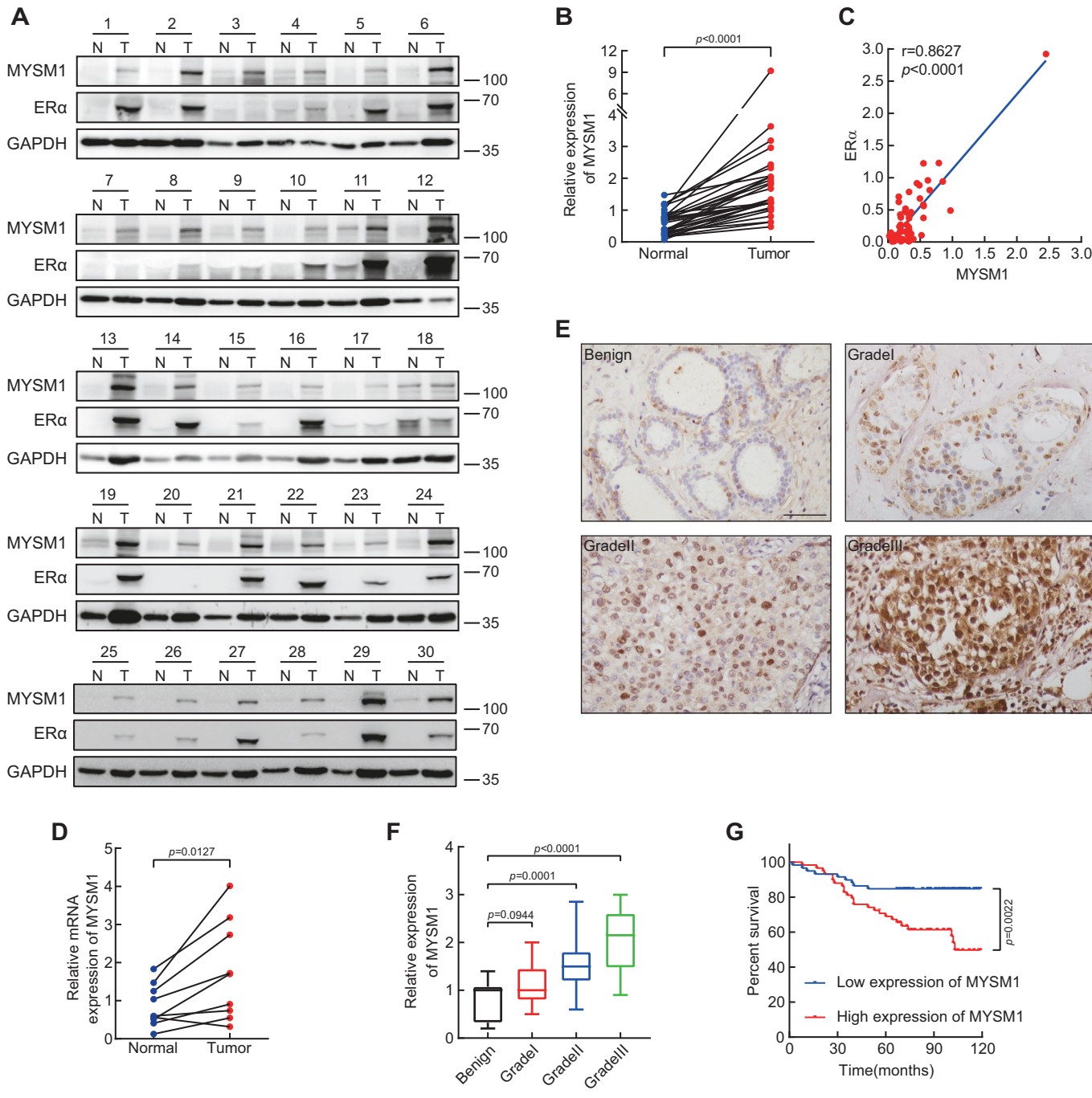

**Figure 7. Expression of MYSM1 is upregulated in clinical Breast Cancer tissues.**

(A) The protein expression of MYSM1 and ERα in 30 presentative pairs of primary BCa (T) and adjacent non-cancerous tissues (N). (B) Grayscale analysis of MYSM1 level was conducted using GAPDH as the internal control and the differential expression of MYSM1 in T versus N was delineated. ***$P < 0.001$ (mean ± SD; Student's $t$ test; $n = 30$). (C) The level of MYSM1 is positively correlated with ERα. Semiquantitative expression of MYSM1 and ERα from the 30-pair samples were statistically analyzed. The relative level of MYSM1 was plotted against that of ERα☐ ***$P < 0.001$ (mean ± SD; Student's $t$ test; $n = 30$). (D) qPCR analysis showing the relative mRNA level of MYSM1 in fresh human breast tissues. *$P < 0.05$ (mean ± SD; Student's $t$ test; $n = 9$). (E) Representative MYSM1 IHC images of primary ERα-positive breast tumors in different grades. Scale bars, 100 μm. (F) Statistical quantifications of MYSM1 expression in benign tissues ($n = 8$) and different grades of breast cancer tissues (grade I $n = 21$, grade II $n = 66$, grade III $n = 54$). The central mark indicates the median, and the bottom and top edges of the box indicate the 25th and 75th percentiles, respectively. ***$P < 0.001$ and N.S. stood for no significant (mean ± SD; Student's $t$ test). (G) Higher MYSM1 level predicts a poor clinical outcome in BCa cases (n = 141). Median expression score of MYSM1 was used as cutoff to evaluate the overall survival by Kaplan–Meier (KM) method. **$P < 0.01$ (mean ± SD; Student's $t$ test). Source data are available online for this figure.

**Table 2. Relationship between MYSM1 expression and clinical pathologic features in breast cancer.**

| Characteristics | Cases (N = 117) | MYSM1 expression | | P value |
| --- | --- | --- | --- | --- |
| | | Low (N = 59) | High (N = 58) | |
| Age | | | | |
| ≤55 | 49 | 24 | 25 | 0.790 |
| >55 | 68 | 35 | 33 | |
| T | | | | |
| T1 | 33 | 18 | 15 | 0.577 |
| T2+ | 84 | 41 | 43 | |
| N | | | | |
| N0/N1 | 91 | 45 | 46 | 0.693 |
| N2+ | 26 | 14 | 12 | |
| Histological grade | | | | |
| I | 5 | 3 | 2 | 0.044 |
| II | 59 | 36 | 23 | |
| III | 53 | 20 | 33 | |
| ERα status | | | | |
| ERα− | 45 | 31 | 14 | 0.002 |
| ERα+ | 72 | 28 | 44 | |
| PR status | | | | |
| PR− | 67 | 35 | 32 | 0.650 |
| PR+ | 50 | 24 | 26 | |
| HER2 status | | | | |
| HER2− | 4 | 4 | 0 | 0.044 |
| HER2+ | 113 | 55 | 58 | |

## Cell lines and culture conditions

All cell lines were obtained from the ATCC (American Type Culture Collection) and authenticated by STR profiling. They were tested mycoplasma-negative prior to experiments. MCF-7 (ATCC: HTB-22) and HEK293 (ATCC: CRL-1573) were cultured in Dulbecco's Modified Eagle Medium (DMEM) medium (Gibco), supplemented with 10% fetal bovine serum (FBS) (Gibco) and 100 U/ml penicillin–streptomycin (P/S). T47D (ATCC: HTB-133) and BT474 (ATCC: HTB-20) were cultured in RPMI-1640 medium (Gibco), supplemented with 10% FBS and 100 U/ml P/S. For estrogen-starving conditions, cells were grown in phenol red-free DMEM or RPMI-1640 containing 5% charcoal-treated serum. All cells were maintained in a humidified incubator at 37 °C and 5% $CO_2$. The Tamoxifen-resistant MCF-7 (MCF-7 TMR) and T47D (T47D TMR) cell lines were gifts from Prof. Xiao (School of Pharmacy, China Medical University) and were cultured in phenol red-free DMEM or RPMI-1640 containing 10% FBS and 1 μM Tamoxifen (Abmole, Cat# M7353).

## siRNA and lentivirus

siRNA control (siCtrl) and siRNA against the gene encoding MYSM1 were purchased from Sigma Aldrich. The RNAi nucleotides were transiently transfected in cells using jet-PRIME transfection reagent (Polyplus) following the manufacturer's instructions. Sequence of siMYSM1: 5'-CAAAUGCGGUCUG-GAUAAAdTdT-3'. Sequence of siCtrl: 5'-UUCUCCGAACGUGU-CACGUdTdT-3'. For lentivirus-delivered RNAi, negative control (shCtrl) and shRNA against MYSM1 (shMYSM1) lentivirus targeting the same sequence as siMYSM1 as above were purchased from Shanghai GeneChem Company. Two days after lentivirus infection, puromycin was added into the medium at a concentration of 3 μg/ml to select stably transduced cells.

## Western blot and co-immunoprecipitation (Co-IP)

Western blot was performed by the standard process as previously described (Zhang et al, 2022). Briefly, samples were lysed in lysis buffer containing protease inhibitor and the supernatants harvested from centrifugation were boiled for denaturation, and resolved by SDS-PAGE. After electrophoresis, proteins were transferred to PVDF membrane, probed with primary antibodies overnight at 4 °C, and incubated with the appropriate secondary antibody for 1 h at room temperature. The respective protein bands were visualized by ECL solution. All data were represented from at least three independent experiments. The antibodies used in this experiment were: anti-MYSM1 (Abcam, cat# ab193081, 1:1000), anti-ERα (Cell Signaling Technology, cat# 8644, 1:1000), anti-c-Myc (Proteintech, cat# 10828, 1:2000), anti-VEGF (Proteintech, cat# 19003-1-AP, 1:1000), anti-Cyclin D1 (Cell Signaling Technology, cat# 2978, 1:1000), anti-GAPDH (ABclonal, cat# AC036, 1:5000), anti-FLAG, anti-His and anti-HA (GNI, 1:1000), anti-p300 (Cell Signaling Technology, cat# 54062, 1:1000), anti-CBP (Cell Signaling Technology, cat# 7389, 1:1000), anti-pCAF (Cell Signaling Technology, cat# 3378, 1:1000).

For immunoprecipitation, experiments were performed on the basis of our previous study (Sun et al, 2020). Supernatant cell lysates were incubated with Protein A/G agarose gel (GE Healthcare) and primary antibodies under constant rotation overnight, then the mixture was rinsed three times before analyzed by immunoblotting.

## Ubiquitination assay

The plasmids were transiently transfected into MCF7 or HEK293 cells for 48 h, incubated with 10 μM MG132 for 6 h. The cells were collected and lysed in the denature lysis buffer. The His/HA-ubiquitinated ERα protein was purified and immunoblotted with anti-His or anti-HA antibodies. HA-tagged different ubiquitin mutants, including K0 (lysine less), K48 (only K48-linked-Ub), and K63 (only K63-linked-Ub), were used in this study as indicated.

## GST-pull down

*E. coli* cells expressing GST-conjugated ERα-AF1 and ERα-AF2 protein were lysed to get the purified protein. Glutathione-Sepharose beads (GE Healthcare) coupled with either GST or with the GST fusion protein GST ERα-AF1 (29-180aa) and GST ERα-AF2 (282-595aa) were incubated with the in vitro transcribed and translated FLAG-MYSM1. After rinsing the beads three times, the proteins bound to the beads were detected by western blot and stained using using Coomassie Brilliant Blue R-250.

## Immunofluorescence (IF)

Cells were fixed at room temperature with 4% paraformaldehyde for 20 min, permeabilized in PBS containing 0.05% Triton X-100 for 20 min, blocked with 1% donkey serum albumin, and incubated with primary antibodies (1:100 dilution) in a humid chamber overnight at 4 °C. After three times washing, specimens were incubated with Alexa Fluor-conjugated secondary antibody (Jackson ImmunoResearch Laboratories Inc) for 1 h followed by DAPI (Roche) staining for 30 min at room temperature and then subjected to confocal microscopy.

## Luciferase dual-reporter assays

Cells were serum-starved overnight and co-transfected with wild-type or mutant MYSM1 expression constructs, along with ERα, ERE-tk-Luc, and an internal control plasmid of Renilla luciferase (pRL). After 4 and 20 h, E2 (100 nM) stimulus was given to corresponding groups respectively to ensure the continuous supply. 24 h after transiently transfection, cells were lysed for luciferase activity detection by a Promega dual-luciferase reporter assay system. Relative luciferase activity as shown in figure is averaged over at least three times.

## RNA isolation and quantitative real-time PCR (qPCR)

MCF-7 and T47D cells cultured in medium containing 5% charcoal-treated serum were transfected with siCtrl or siMYSM1. Sixteen to 18 h later, cells were treated with 100 nM E2 or equal ethanol for 6 h before harvest. Total RNA was extracted with RNA Trizol (TAKARA) following the manufacturer's recommendations. One μg of total RNA was reversely transcribed into cDNA using the PimeScript RT-PCR kit (TAKARA) and analyzed by qPCR on LightCycler96 system (Roche) with the SYBR premeraseTaq kit (TAKARA). Amplified mRNA levels were normalized to 18s mRNA. Primers used were listed in Table EV2. All data were represented from at least three independent experiments.

## Chromatin immunoprecipitation (ChIP) and ChIP re-IP

Cells transfected with siCtrl or siMYSM1 were cultured in phenol red-free medium with 5% charcoal-treated serum for 2 days followed by 4 h E2 (100 nM) or equal EtOH stimulation. Cells were cross-linked with 1% formaldehyde at room temperature before glycine (0.25 M) quenching. After cell collection, lysis buffer was added and the mixture was sonicated on ice. Sheared-chromatin solutions were separated by centrifugation at 12,000 rpm 15 min at 4 °C and an eighth of the supernatant was kept for input. Concurrently, the remaining supernatant was immunoprecipitated using specific antibodies overnight. After that, specimens were incubated with protein A-sepharose beads for 4 h on a mixing platform before sequentially beads-washing by low salt buffer, high salt buffer, LiCl buffer, and TE buffer. The protein-DNA complexes were eluted by elution buffer followed by crosslink reversal and proteinase K digestion. Finally, the purified DNA was precipitated in absolute alcohol and resuspended in TE buffer. qPCR was performed to examine the precipitated genomic DNA samples (Nelson et al, 2006). All figures about ChIP represented the results of three independent experiments. The sequences of the primers described in Table EV3. The antibodies used in this experiment were: anti-MYSM1 (Abcam, cat# ab193081, 1:100), anti-ERα (Cell Signaling Technology, cat# 8644, 1:100), anti-p300 (Cell Signaling Technology, cat# 54062, 1:50), anti-CBP (Cell Signaling Technology, cat# 7389, 1:50), anti-pCAF (Cell Signaling Technology, cat# 3378, 1:25), anti-H2AK119ub (Cell Signaling Technology, cat# 8240, 1:100), anti-H2BK120ub (Cell Signaling Technology, cat# 5546, 1:200), anti-H3K4me3 (Sigma-Aldrich, cat# 07-473, 1:100), anti-H3K9Ac (Cell Signaling Technology, cat# 9649, 1:50), anti-H3K27Ac (Cell Signaling Technology, cat# 8173, 1:100).

For ChIP re-IP, the cross-linked immunocomplex was eluted from the first ChIP by 10 mM dithiothreitol incubation at 37 °C for 30 min and diluted 50-fold in re-ChIP buffer. The products were subjected to the ChIP procedure again with the indicated antibodies.

## Micrococcal nuclease (MNase) assays

MNase assays were conducted following the protocol as described previously (Peng et al, 2009). After MYSM1-depleted and MYSM1-overexpression MCF-7 cells harvest, nuclei were extracted and digested with gradually increased concentrations of MNase (0, 20, 40, 80 U/ml) in the digestion buffer (15 mM Tris-HCl, pH7.4, 15 mM NaCl, 60 mM KCl, 1 mM $CaCl_2$, 0.25 M sucrose, and 0.5 mM DTT) at 37 °C for 20 min. The digested genomic DNA was carefully purified and subjected to 1.2% agarose gel electrophoresis.

## Cell viability, colony formation, and flow cytometric analysis

Cells infected with shCtrl or shMYSM1 lentivirus were seeded in 96-well plates at 3000 cells per well, in triplicate, complemented with estradiol, Tamoxifen (Abmole, Cat# M7353), Fulvestrant (Abmole, Cat# M1966) or Letrozole (Abmole, Cat# M3699) for 7 days. Viable cells at different time points were measured by MTS assay (Promega, Cat# G3580) with the absorbance at a wavelength of 490 nm.

For colony formation assay, cells were planted in six-well plates and cultured in medium with E2, Tamoxifen, Fulvestrant or Letrozole treatments at stated concentrations for 2 weeks, then were fixed by 4% paraformaldehyde for 15 min before undergoing crystal violet staining of colony number count.

For analysis of cell cycle, cells were harvested with EDTA-free trypsin 2 days after E2 (100 nM) addition and fixed with 75% ethanol at −20 °C overnight. After ice-cold PBS washing for once the next day, nuclear DNA was stained by propidium iodide for 15 min in a dark room and determined on a flow cytometer (Becton Dickinson).

## Mouse models and xenograft studies

Four-week-old female *BALB/c* nude mice (Vital River Laboratory) were housed in ventilated cages under specific pathogen-free (SPF) conditions at the Model Animal Research Center of China Medical University. All animal procedures were approved and supervised by the Institutional Animal Care and Use Committee of (IACUC) of China Medical University, with protocol approval number

## The paper explained

### Problem

The most common subtype of diagnosed breast cancer (BCa) is estrogen receptor α (ERα) positive. While targeting ERα with endocrine therapy is the current standard of care for this subtype, endocrine resistance remains a major issue leading to recurrence and metastasis. As a key transcriptional factor, ERα action is deeply influenced by its co-regulators. Thus, the discovery of novel co-regulators may help elucidating the potential mechanisms of ERα pathway and provide potential therapeutic strategies for BCa treatment.

### Results

We report MYSM1 as a novel ERα co-activator that associates with ERα and increases ERα-induced transcriptional activity in BCa through non-histone and histone deubiquitination. As a deubiquitinase, MYSM1 stabilizes ERα protein through its MPN enzyme catalytic domain and is recruited with ERα at the promoters of E2-induced genes, thereby triggering transcription initiation and subsequent acceleration in cell proliferation and suppression of antiestrogen sensitivity.

### Impact

Through multi-level studies of cell culture, pathological specimens and animal models, we identify a novel ERα co-regulator and provide mechanistic insights of its role in development and endocrine resistance of ERα-positive BCa. These findings suggest that MYSM1 could be a potential therapeutic target in ERα-positive BCa diagnosis and treatment.

#TZ2020074. In all, $5 \times 10^6$ MCF-7 cells stably expressed shCtrl and shMYSM1 lentivirus were suspended in 100 μl of culture medium/Matrigel (1:1) (BD Biosciences) and implanted subcutaneously into the bilateral armpits of *BALB/c* nude mice. Tumor diameter was measured every 5 days and tumor volume was calculated as follows: $V = (0.5 \times \text{width}^2 \times \text{length})$ mm$^3$. Four weeks after inoculation, tumor-bearing mice were sacrificed following the policy for the humane treatment of animals and tumor tissues were isolated, weighing, followed by formalin-fixing and paraffin-embedding for IHC staining.

## Virtual screening and molecular docking

Drug candidates were retrieved from the ZINC database to create the ligand library. Chemoinformatics analysis and filtering of compounds were done using the program MOE (Chemical Computing Group, Canada) with the crystal structure of MYSM1 protein (AlphaFold ID: AF-Q5VVJ2-F1) following the virtual screening workflow. The binding energies of the screened compounds were given in kcal/mol, with the larger the negative number, the better the binding effect. PyMOL molecular graphics system (https://pymol.org/2/) was used for molecular docking to visualize the docking position.

## Patients and tumor specimens

Human primary BCa tissues and corresponding adjacent tissues were obtained from Liaoning Cancer Hospital and the First Affiliated Hospital of China Medical University. All specimens were collected with patients' informed consents and ethics approval for the study was obtained at the Institutional Review Board of China Medical University.

The experiments conformed to the principles set out in the WMA Declaration of Helsinki and the Department of Health and Human Services Belmont Report.

## Immunohistochemical (IHC) analysis

Commercial BCa tissue microarray slides (HBre-Duc150Sur-02) were purchased from Shanghai Outdo Biotech Co., Ltd. (SOBC). Paraffin-embedded tissue sections were deparaffinized and rehydrated before endogenous peroxidase removal by 3% hydrogen peroxide for 15 min. Subsequently, tissues were boiled in citrate buffer (pH 6.0) for antigen retrieval in a pressure cooker followed by protein block with goat serum for 30 min at room temperature. The sections were next incubated with anti-MYSM1 (Sigma-Aldrich, cat# HPA054291, 1:1000), anti-ERα (Cell Signaling Technology, cat# 8644, 1:200), c-Myc (Proteintech, cat# 10828, 1:400), anti-Ki67 (Cell Signaling Technology, cat# 9129, 1:400).

Antibodies respectively at 4 °C overnight and with the secondary horseradish peroxidase-labeled polymer anti-rabbit IgG at room temperature for 20 min. Using DAB as a chromogen, the nuclei were counter stained with hematoxylin. Slides were scanned with an Olympus microscope at ×20. The staining scores were evaluated by *H* score method based on the proportion of positively stained tumor cells (0–100%) and the brown intensity (0–3). The final expression score, which ranged from 0 to 3, was determined by multiplying these two independent indexes.

## Statistics

Statistical analyses were performed using GraphPad Prism 8.0. and the SPSS statistical software program. For all in vitro and in vivo experiments, two-tailed Student's *t* test was performed to calculate the *P* value and data in bar graphs represent mean ± SD of at least three biological replicates. For animal experiments, mice were randomized in an unbiased fashion. Researchers were not blinded during mouse experiments. Sample sizes were selected to give a 90% chance of observing statistically significant deviations at $P < 0.05$ in efficacy between the shCtrl and the shMYSM1 group. All the experimental units treated were included in the analysis. Correlation between MYSM1 expression and clinical parameters was determined by chi-square test. Overall survival curves were plotted according to the Kaplan–Meier method with the log-rank test applied for comparison (Chen et al, 2018).

# Data availability

This study includes no data deposited in external repositories.

# Peer review information

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

## Acknowledgements

We appreciate Dr Yunlong Huo for helpful technique assistance. We thank Dr. Shigeaki Kato (Soma Central Hospital, Fukushima, Japan) for giving us the valuable suggestion and comments for the whole work, and providing a pERE-tk-Luc reporter vector, expression plasmids for ERα, and its truncated mutants. We thank Dr Yujie Sun (Nanjing Medical University, Nanjing, China) for MCF-7

cells, which were purchased from American Type Culture Collection. This work was supported by National Natural Science Foundation of China (32370634, 32170603, 31871286 for YZ, 81872015, 82273123 for CW, 32100440 for GS); China Postdoctoral Science Foundation (276066) for GS; Foundation of Liaoning Province of China (LJKZ0756 for SW); local projects supported by the central government (2022JH6/100100035 for YZ); foreign expert project of Ministry of Science and Technology (G2022006007L for YZ).

## Author contributions

**Ruina Luan**: Conceptualization; Data curation; Formal analysis; Validation; Investigation; Methodology; Writing—original draft; Writing—review and editing. **Mingcong He**: Software; Funding acquisition; Project administration. **Hao Li**: Data curation; Formal analysis; Validation. **Yu Bai**: Resources; Software; Visualization. **Anqi Wang**: Resources; Supervision; Investigation. **Ge Sun**: Supervision; Funding acquisition; Investigation. **Baosheng Zhou**: Software; Funding acquisition; Methodology. **Manlin Wang**: Supervision; Validation; Methodology. **Chunyu Wang**: Resources; Supervision; Project administration. **Shengli Wang**: Conceptualization; Resources; Funding acquisition. **Kai Zeng**: Resources; Funding acquisition; Visualization. **Jianwei Feng**: Resources; Project administration. **Lin Lin**: Supervision; Project administration. **Yuntao Wei**: Resources; Validation. **Shigeaki Kato**: Visualization; Project administration. **Qiang Zhang**: Resources; Validation. **Yue Zhao**: Writing—review and editing.

## Disclosure and competing interests statement

The authors declare no competing interests.

# Expanded View Figures

**Figure EV1.  MYSM1 interacts with ERα in breast cancer cells.**

(**A**) Evaluation of GFP and ERα level in F1 progeny flies with CG4751 loss of function (lane 3) or MYSM1 gain of function (lane 4) mutants. The lower panels represent merge images. (**B**) The MYSM1-related protein-protein interaction (PPI) networks generated by the STRING online database. (**C, D**) Co-immunoprecipitation conducted in T47D cells to detect the association between endogenous MYSM1 and ERα in response to E2 treatment. (**E**) HEK293 cells complemented with ERα or the deletion mutants of MYSM1 were lysed. Complexes precipitated by anti-FLAG were purified and immunoblotted with indicated antibodies. (**F**) Immunofluorescent staining of MYSM1 (anti-FLAG, green) and ERα (anti-ERα, red) in HEK293 cells overexpressing MYSM1 truncated mutants and ERα in the presence of E2. Nuclei were stained with DAPI (blue). Scale bars, 10 μm. Data information: (**C–F**): $n = 3$ independent experiments performed in duplicate.

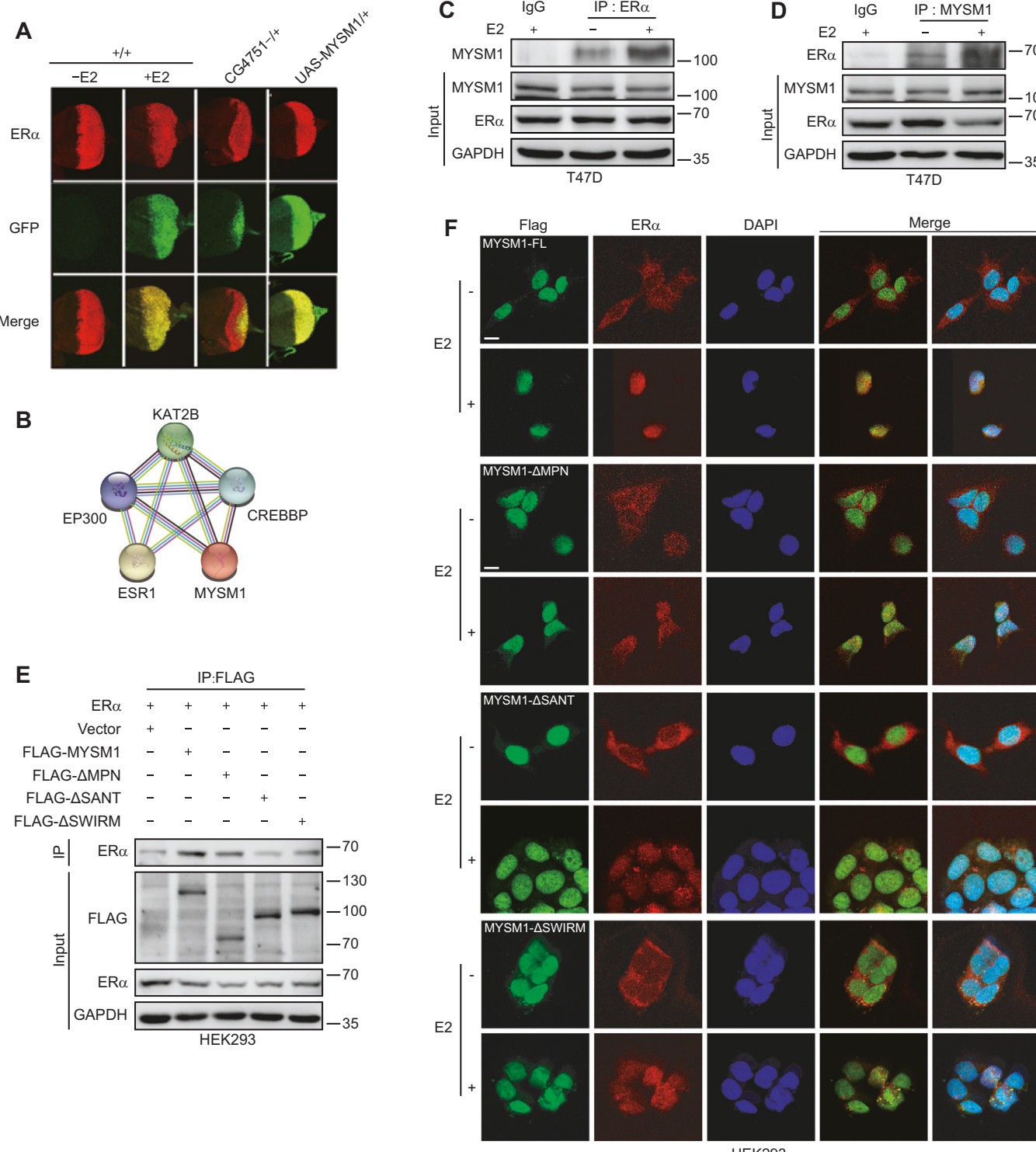

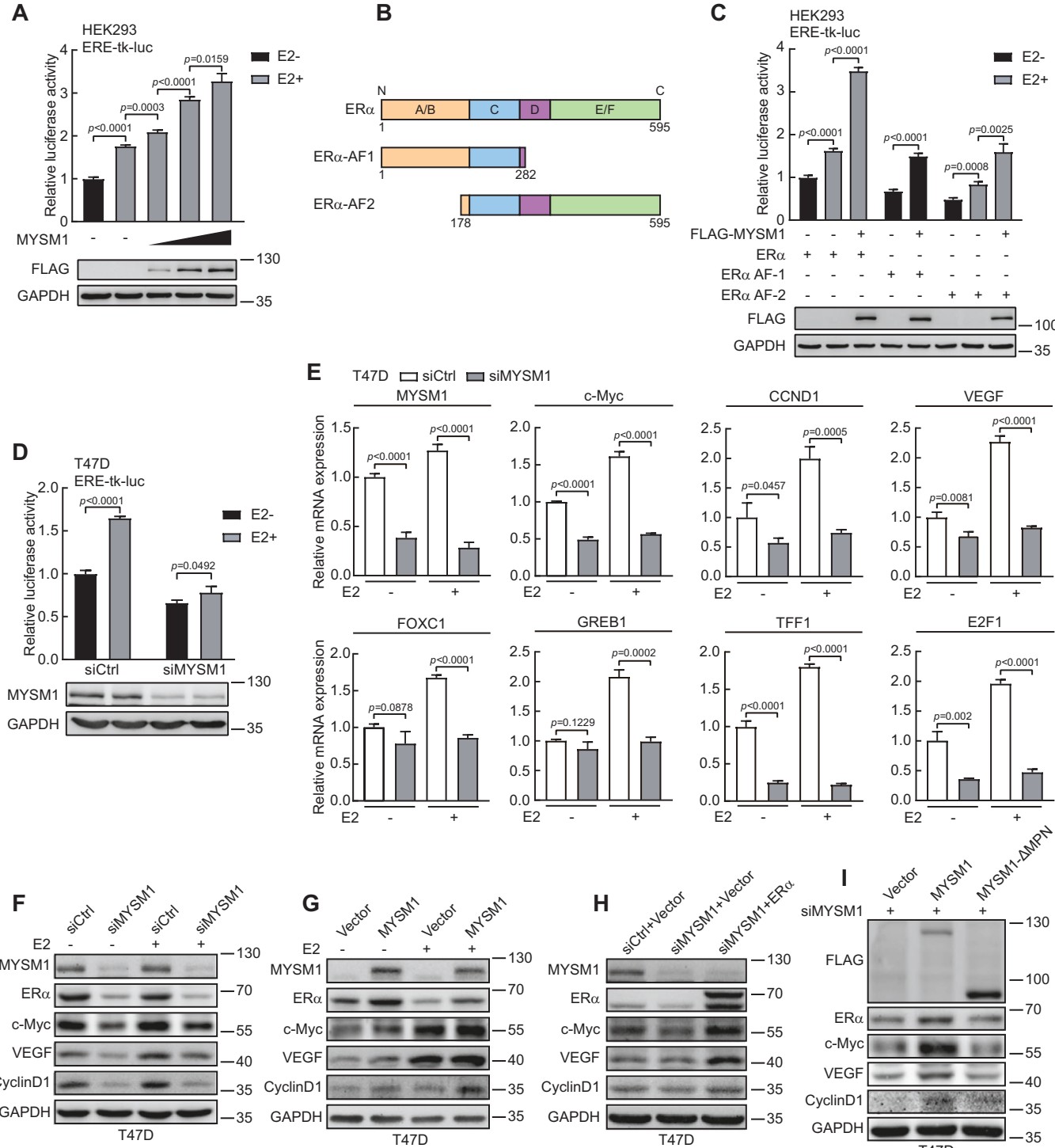

◀ **Figure EV2.  MYSM1 enhances ERα-mediated gene transcription in mammalian cells.**

(A) MYSM1 stimulates ERα-mediated gene transcription in a dose-dependent manner. HEK293 cells were transfected with gradually increased amount of ectopic MYSM1 (0.05 μg, 0.1 μg, or 0.2 μg respectively). MYSM1 expression was examined with anti-FLAG by western blot. (B) Schematic representation of ERα, ERα-AF1, and ERα-AF2 plasmids used in luciferase reporter assays. (C) Relative luciferase activities in HEK293 cells transfected with ERα full length or truncated mutants harboring ERα AF-1 or ERα AF-2 together with MYSM1 expression plasmid in the presence or absence of E2 (100 nM). The expression of MYSM1 was detected by western blot. (D) Effect of MYSM1 knockdown on ERα-induced transactivation. The relative luciferase values in T47D cells were examined after transient transfection of siCtrl or siMYSM1 followed by ERα expression plasmid. (E) mRNA levels of several ERα target genes in T47D cells with MYSM1-depleted. (F, G) Immunoblot of ERα target gene expression using the indicated antibodies in MYSM1-depleted T47D cells (F) and MYSM1-overexpressed T47D cells (G) with or without E2 (100 nM) treatment for 16-18 h. (H) The loss of ERα and its target genes in MYSM1-depleted T47D cells can be rescued by ectopic ERα expression. T47D cells were transfected with siCtrl or siMYSM1 followed by PcDNA3.1/ERα expression plasmid. (I) Western blot detecting the protein levels of ERα and its target genes in MYSM1-depleted T47D cells transfected with PcDNA3.1/MYSM1/MYSM1-ΔMPN expression plasmids. Data information: *$P < 0.05$, **$P < 0.01$, ***$P < 0.001$ (mean ± SD; Student's *t* test; $n = 3$ independent experiments).

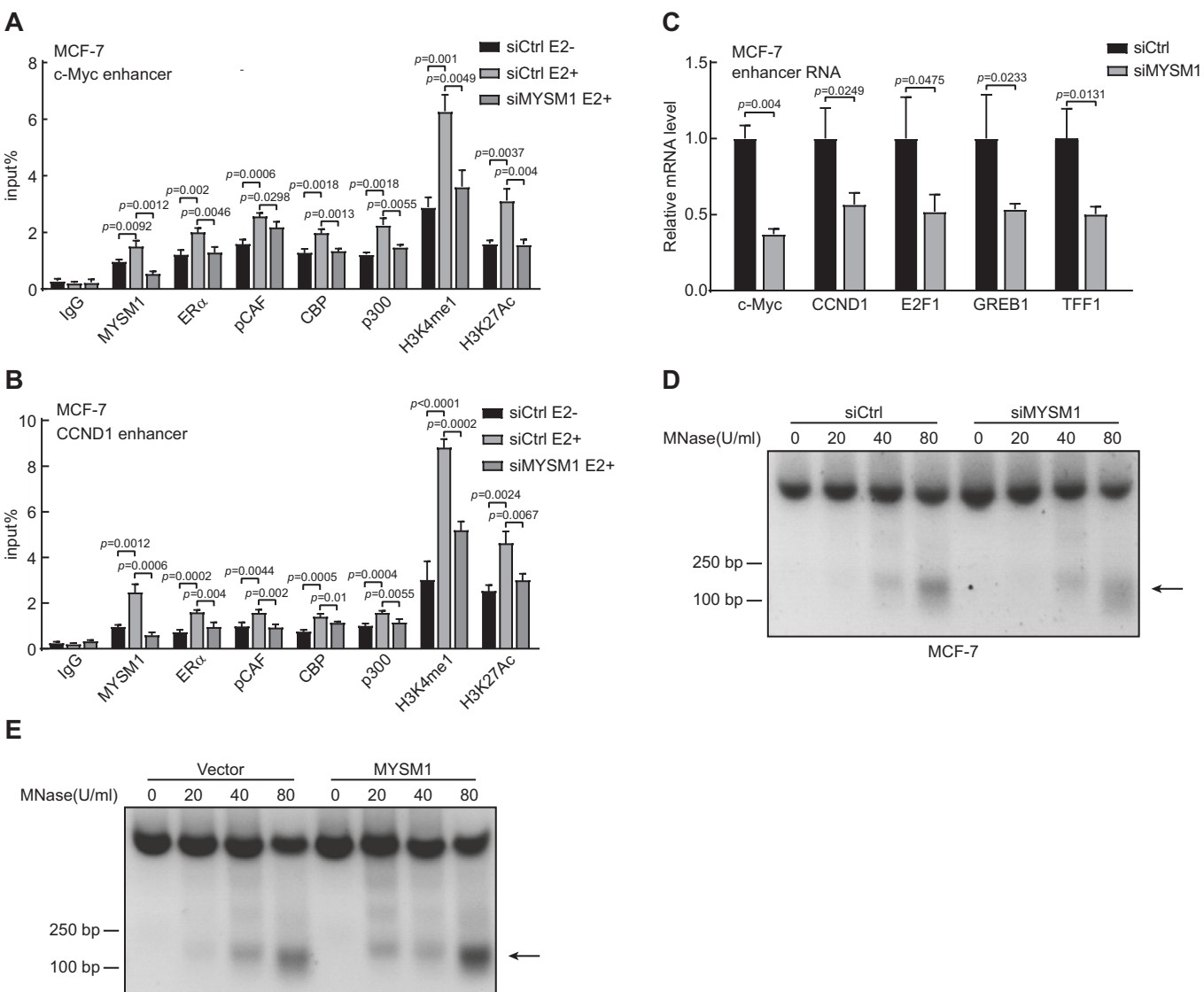

**Figure EV3. MYSM1 enhances the occupation of ERα and HAT complex on E2-regulated enhancers in MCF-7 cells.**

(A, B) ChIP assays via designated antibodies demonstrating the recruitment of MYSM1, ERα, pCAF, CBP, p300 and the histone modification levels H3K4me1 and H3K27ac at the ERα binding site on *c-Myc* (A) or *CCND1* (B) enhancer in MCF-7 cells upon MYSM1 depletion. (C) qPCR analysis demonstrating the eRNA expression of estrogen-induced genes in MCF-7 cells transfected with siCtrl or siMYSM1 in the presence of E2. (D, E) MNase experiments examining chromatin accessibility after digestion with 0, 20, 40, or 80 U MNase in MYSM1-depletion (D) or MYSM1-overexpression (E) MCF-7 cells. Data information: (A–C): $n = 3$ independent experiments. *$P < 0.05$, **$P < 0.01$, ***$P < 0.001$, N.S. means no significance (mean ± SD; Student's $t$ test).

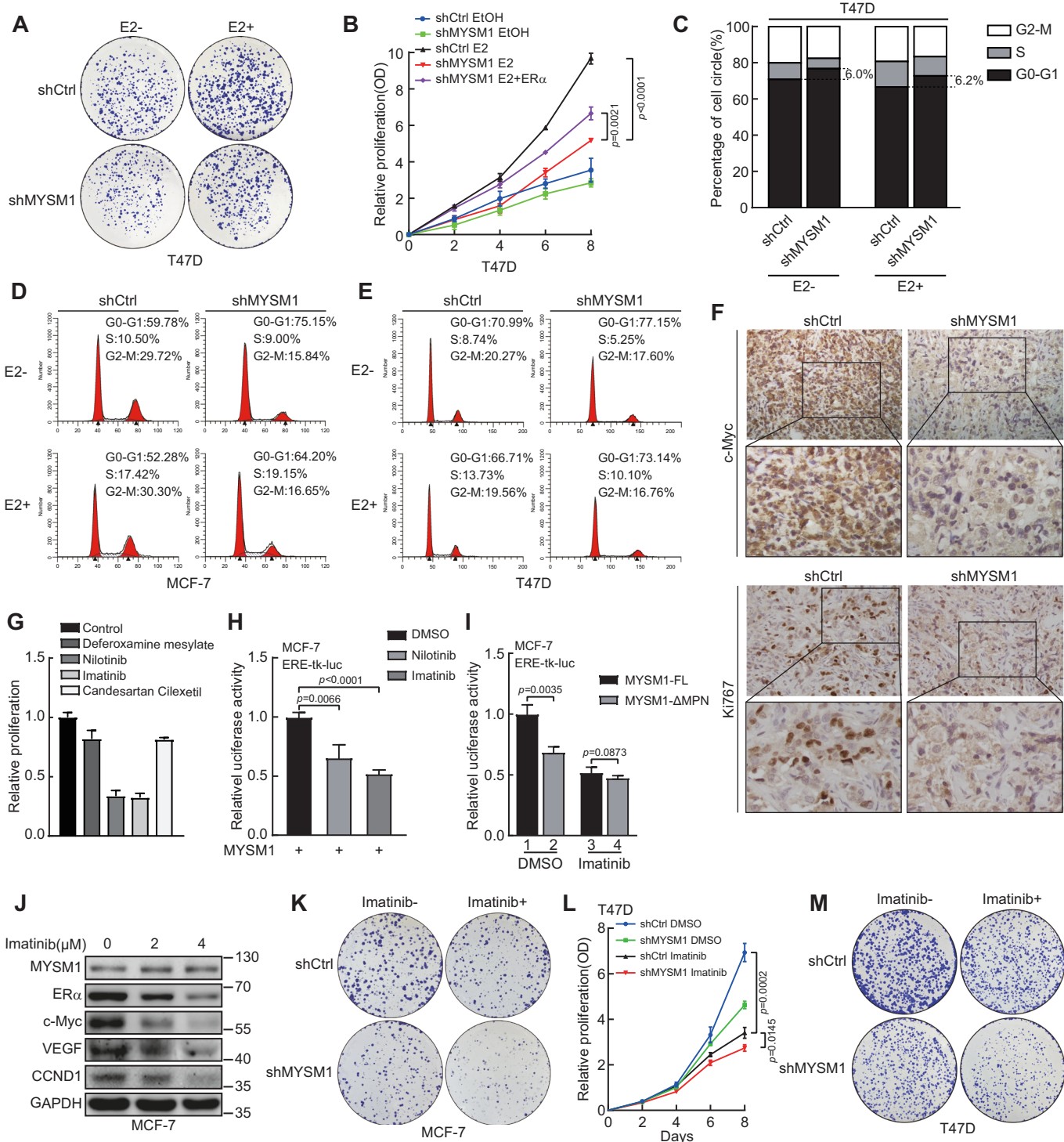

◀  **Figure EV4.  MYSM1 depletion or the small molecule Imatinib docked to MYSM1 suppresses breast cancer cell growth through MYSM1-ERα axis.**

(A) Influence of MYSM1 deficiency on T47D cells as illustrated by colony formation. (B) Relative proliferation rates of T47D cells carrying shCtrl or shMYSM1 with or without E2 (100 nM) by MTS assay. (C–E) Flow cytometric analysis of the cell cycle for MCF-7 (D) and T47D (E) with MYSM1 depletion. All figures about flow cytometry represented the results of three independent experiments. The proportion of the T47D cell population in different phases are listed in (C). (F) c-Myc and Ki67 expression by IHC analysis in xenograft tumors derived from MCF-7 (shCtrl) group and MCF-7 (shMYSM1) group. (G) Histograms showing the effects of Deferoxamine mesylate (2 μM), Nilotinib (2 μM), Imatinib (2 μM), or Candesartan Cilexetil (2 μM) on MCF-7 cell proliferation. The results are expressed relative to the control (2μM DMSO treated). (H) The effect of Nilotinib (2 μM) or Imatinib (2 μM) on luciferase activity in MCF-7 cells transfected with ERα-related dual-luciferase reporter system and MYSM1 expression plasmid. (I) The effect of Imatinib (2 μM) on luciferase activity in MCF-7 cells transfected with ERα-related dual-luciferase reporter system, MYSM1-FL or MYSM1-ΔMPN expression plasmid. (J) Western blot showing the expression of the indicated proteins in MCF-7 cells treated with different concentrations of Imatinib (0 μM, 2 μM, or 4 μM respectively). (K) Colony formation assay showing the effect of Imatinib treatment (1 μM) on shCtrl or shMYSM1 stably expressed MCF-7 cells. (L) Growth curve showing the effect of shMYSM1, Imatinib, or their combination (1 μM) on T47D cell proliferation. Total cell viability was assessed every other day by MTS assay. (M) Colony formation assay showing the effect of Imatinib treatment (1 μM) on shCtrl or shMYSM1 stably expressed T47D cells. Data information: ***P* < 0.01, ****P* < 0.001 (mean ± SD; Student's *t* test; *n* = 3 independent experiments).

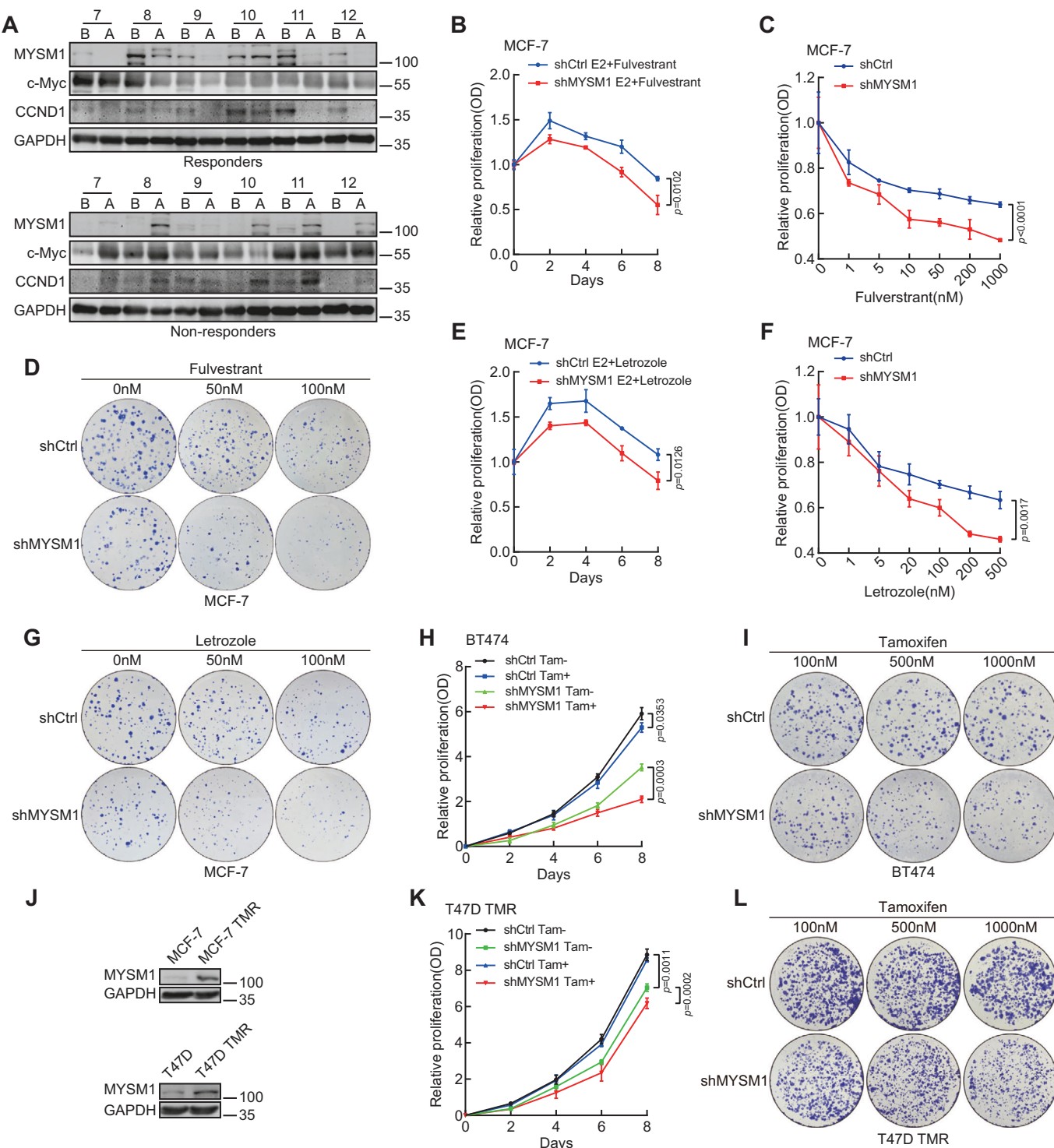

◀ **Figure EV5. MYSM1 depletion subjects ERα-positive breast cancer cells to endocrine treatment.**

(A) MYSM1 and CCND1 protein expression in the "Responders" and "Non-responders" samples before and after AI treatment were examined by western blot. "B" represents cases before AI treatment, "A" represents cases after AI treatment ($n = 6$). (B, E) Growth curve showing the effect of MYSM1 knockdown on MCF-7 cell proliferation with Fulverstrant (200 nM) (B) or Letrozole (100 nM) (E) treatment. Total cell viability was assessed every other day by MTS assay. (C, F) A cellular viability detection in MYSM1-deletion MCF-7 cells that incubated in various concentrations of Fulverstrant (C) or Letrozole (F) for 7 days. (D, G) MCF-7 cells with/without MYSM1 knockdown were subjected to colony formation assay under diverse doses of Fulverstrant (D) or Letrozole (G). Clones were stained with R250 and photographed 2 weeks later. (H, K) The line chart renders the relative proliferation of the shMYSM1 group compared to the shCtrl group in BT474 (H) or T47D TMR (K) cells in the stimulation of Tamoxifen (1 μM) or not. (I, L) The panels show colony-formation assay conducted in BT474 (I) or T47D TMR (L) cells infected with lentivirus expressing shCtrl/shMYSM1. Cells in each panel were treated with different doses of Tamoxifen for 15 days before fixation and R250 staining. (J) Western blot assay detecting MYSM1 protein expression in MCF-7, T47D, and their corresponding Tamoxifen-resistant (TMR) cells. Data information: $*P < 0.05$, $**P < 0.01$, $***P < 0.001$ (mean ± SD; Student's $t$ test; $n = 3$ independent experiments).

