## [Peer Review File · EMBO Molecular Medicine]

MYSM1 acts as a novel co-activator of ER α to confer antiestrogen resistance in breast cancer

Ruina Luan, Mingcong He, Hao Li, Yu Bai, Anqi Wang, Ge Sun, Baosheng Zhou, Manlin Wang, Chunyu Wang, Shengli Wang, Kai Zeng, Jianwei Feng, Lin Lin, Yuntao Wei, Shigeaki Kato, Qiang Zhang, and Yue Zhao

DOI: [10.15252/emmm.202317672](https://doi.org/10.15252/emmm.202317672)

Corresponding authors: Yue Zhao (yzhao30@cmu.edu.cn) , Qiang Zhang (zhangqiang8220@163.com)

Review Timeline:

Transfer Date from Review Commons:	7th Mar 23
Editorial Decision:	8th Mar 23
Revision Received:	19th Aug 23
Editorial Decision:	21st Sep 23
Revision Received:	26th Oct 23
Accepted:	6th Nov 23

Editor: Lise Roth

Transaction Report:

This manuscript was transferred to EMBO Molecular Medicine following peer review at Review Commons.

Review #1

1. Evidence, reproducibility and clarity:

Evidence, reproducibility and clarity (Required)

Summary:

ER+ breast cancer is the most common form of cancer. Targeting ER-alpha transcriptional cofactors present one potential method to target the disease. The authors demonstrate that MYSM1 is a histone deubiquitinase and a novel ER cofactor, functioning by up-regulating ER action via histone deubiquitination.

Loss of MYSM1 attenuated cell growth and increase breast cancer cell lines' sensitivity to anti-estrogens. The authors, therefore, propose MYSM1 as a potential therapeutic target for endocrine resistance in Breast cancer.

Major Comments:

- The data as presented is convincing, and the evidence for the role of MYSM1 as a co-activator of ER-alpha is extensive.
- Given the amount of data, I do not believe any additional experiments are needed.
- I could not find any description of ethics for the patient samples used.

Minor

- Figure 4C - is the increase of binding in response to Estrogen significant? It is an important control to show for MCF7 as Fig 4B is in T47D.
- Figure 6 - Can we clarify that B = Before, A = After
- The use of Fig EV was confusing to me, I assume it means supplementary?

2. Significance:

Significance (Required)

- Discovery science to understand the regulation of the ER is critical in discovering new opportunities to target breast cancer. As far as I can tell this is the first study where MYSM1 is a co-regulator of the ER.
- The significance would be greatly increased if the manuscript identified opportunities to target the ER via this pathway using existing compounds. However, it is reasonable to consider this is beyond the scope of this study.

3. How much time do you estimate the authors will need to complete the suggested revisions:

Estimated time to Complete Revisions (Required)

(Decision Recommendation)

Less than 1 month

Yes

Review #2

1. Evidence, reproducibility and clarity:

Evidence, reproducibility and clarity (Required)

Below I outline a few suggestions that can help clarify specific aspects of the study.

Fig. 2: Ideally a rescue study with a wild-type and catalytically mutant MYSM1 should be performed.

What is the ERa interactome in the presence and absence of MYSM1? Proteomics studies upon shMYSM1 should be performed. Alternatively, a better characterization of ERa-containing complexes upon shMYSM1 should be performed.

Fig. 3: Does MYSM1 control its own protein via deubiquitination?

Fig. 4: I propose that the authors perform MYSM1 ChIP-Seq to better show the MYSM1

distribution and overlap with ERa distribution.

Fig. 7. Is there a correlation between MYSM1 mRNA and protein levels in cancer and physiological samples? How is the MYSM1 transcriptionally regulated in physiological and cancer cells?

2. Significance:

Significance (Required)

This is a very comprehensive study characterizing the role of MYSM1 deubiquitinase in ERa transcriptional programs in breast cancer systems. Breast cancer therapy is an unmet need and the role of deubiquitinases warrants further investigation. This accounts for the high significance of the story.

3. How much time do you estimate the authors will need to complete the suggested revisions:

Estimated time to Complete Revisions (Required)

(Decision Recommendation)

Between 1 and 3 months

Yes

Review #3

1. Evidence, reproducibility and clarity:

Evidence, reproducibility and clarity (Required)

Luan et al performed a detailed analysis on the potential coactivator MYSM1 and its role in regulating the expression of ER and ER-dependent genes by being a deubiquitinase of ER as well the repressive mark, H2Aub1. This study has demonstrated an excellent work on the biochemistry aspect of the story with meticulous work on the role of specific domains of MYSM1 and ER and how they interact and how the deubiquitination process is regulated. This was identified initially in Drosophila models, but eventually and promptly explored in multiple breast cancer cell lines and patient samples.

****Major Comments:****

1. It is really exciting to see how MYSM1 regulates ER activity and it looks like expression of ER is the first event of regulation by MYSM1's. However, H3Ac would be the very intermediate event of ER activity. This brings a question of whether ER complex itself is affected by MYSM1 - for example, does MYSM1 affect p300, SWI/SNF and other ER-associated coactivator binding? Does it affect chromatin accessibility? Which exact histone mark of H3Ac is affected, as different proteins are involved in the acetylation of histones.
2. The regulation of MYSM1 is mainly shown on promoters of ER regulated genes. However, ER primarily bind to enhancers. Is there any general effect on enhancers?
3. MYSM1 is not the complex usually cells prefer to deubiquitinate H2Aub, but BAP1. What is the role of BAP1 here? Are they redundant or any cross-talk?
4. Effect of MYSM1 on histone marks on the EREs - only one ERE is shown. Multiple EREs should be validated by qPCR. Enhancers should also be focused. Does it affect H3K27ac or H3K4me1?
5. It is clear that MYSM1 is required for the response to antiestrogen therapies. However, the link to resistance is not completely clear. This should be investigated with multiple Tamoxifen resistant cell lines. There is one cell line used, but it is responding to tamoxifen even at lower concentrations in Crystal violet assays. MYSM1 overexpression in nonresponders doesn't mean that their activity is also more. Binding analyses should be analysed in proper Tamoxifen-resistant cell lines. Usually, Tamoxifen is used or works at concentrations from 100 nM - 1 uM in vitro to see the transcriptional effects. However, the authors claim that these are very high concentrations, but actually they aren't the concentrations which promote toxicity.
6. Discussion about DUB inhibitors - how specific are these? Would they be useful to target MYSM1 activity and thus ER regulation in nonresponders or resistant cell lines? This would add up strongly on the clinical potential of the study.
7. OPTIONAL: ChIP-seq analyses on the factors would be more informative to look at the unbiased mechanisms including enhancers.
8. Number of replicates aren't clear in figure legends. Are they biological or technical replicates?

****Minor comments:****

9. Please give page numbers and line numbers in the manuscript.
10. Title - "MYSM1 co-activates ER action". "Action" is not needed to be mentioned here.
11. Abstract talks about the work on Drosophila mainly, but apart from the first experiment, everything else is done on mammalian cell culture and also clinically relevant patient samples.

12. Abstract Line 13 - the work is done many ER regulated genes and not gene.
13. Pg 6 first paragraph - What/how many mutants were screened here?
14. CoIP protocol is not clear. It says followed with manufacturer instructions but no kit information is provided.
15. Fig. 1H, etc - can you show a zoomed in or DAPI removed (from merge) picture to show the interactions clearly? It's hard to follow the yellow co-interaction spots as they are hidden behind the blue colour. Any kind of quantification analyses would be wonderful.
16. Fig. EV1H - can you link this with the results from Fig. 1F to discuss if the delta SANT-MYSM1 lost the interaction with ER also in the IF studies?
17. Pg 7 - 3-4th line from last - These lines should move above where AF1 and AF2 are introduced. According to Fig. 1G, the interaction of AF2 and MYSM1 is important. Why do we see an effect on AF1 as well in Fig. 2B?
18. It's confusing to have HEK and breast cancer cell line datasets swapped inconsistently between main figures and Supplementary figures. It would be nice to keep them consistent.
19. RPMI is spelled wrong in Pg. 19.
20. How long is the estrogen treatment done in each experiment? What is the concentration? This should be mentioned in the figure legends. 12 or 24 hrs time point is a later stage of estrogen receptor induction. Even 1-3 hrs would be sufficient to promote a stronger effect on RNA transcription than that of these later time points. What you are looking at is all effect on later time points and the effect should be observed on earlier time points to observe dynamic and immediate effects. p-values are required for the comparison on no E2 vs E2 here.
21. Fig. 2G - effect on c-Myc after MYSM1 knockdown is not clear comparing to the previous WB in 2E.
22. Pg. 8 - start of the last paragraph - "Unexpectedly, in Co-IP experiment as shown in Figure 2E and F" - These are not Co-IP experiments.
23. Fig. 3C and E - Quantification with comparison needed.
24. Pg 10 - subtitle - multiple gene promoters have been looked, but the subtitle says "gene". Only ERE for c-MYC is looked at, but it says EREs.
25. MYSM1 is in the nucleus in IF even before E2 treatment, however it is recruited after estrogen treatment in ChIP assays. Explain why there is a difference seen here. What other targets they might bind to in the nucleus?
26. Pg. 10 last line - the sentence should be combined with comma.
27. Fig. 5H - What about Ki67 which is a proliferative marker for cancer cell growth?
28. Pg 12 - Samples were used from patients treated with AI adjuvant treatment. A small summary of details are needed here including n, arm, details of administration, etc even though mentioned in Methods.
29. MYSM1 is upregulated in nonresponders, but it is also downregulated in responders which is ignored. What would this mean mechanistically? Don't patients need MYSM1 for the response or after treatment? Does estrogen inhibition regulate MYSM1 upstream?
30. Pg 13 - Is this data associated with any trial? More details are needed.
31. Last lines of Pg 15 - These were already introduced in the results.
32. Pg 16 - third last line of the first paragraph - makes typo.

2. Significance:

Significance (Required)

- The study seems to be novel as MYSM1 is never studied before as a coactivator for ER. This expands the wealth of knowledge we have on coactivators which can be explored for its potential targeting to treat advanced breast cancers. The study seems to be support the biochemical aspects of ER interaction, but vaguely uncovers the functional or epigenetic mechanisms.
- Studies on coactivators/coregulators of ER is very important, as modulating ER alone is not efficient enough to solve the puzzle of antiestrogen resistance. The expression/activity levels of the coregulators are very important as these can be modulated in cancers due to epigenetic reprogramming during resistance and mutations on these genes dominate. They can also serve as potential targets especially when cells don't respond to classical ER targeting therapies.
- Strength - Biochemical analyses of the interactions and detailed mechanistic information
- Limitation - Studies are very much limited to the biochemical regulation on ER and not on the molecular or epigenetic mechanisms. Association of MYSM1 in resistance mechanisms isn't clear.
- Audience - this can be interesting for both basic research and clinical audience. Biochemical knowledge would help people to understand how a nonclassical deubiquitinase can promote nuclear receptor associated transcription by targeting genomic and nongenomic targets simultaneously. Clinically this study would be relevant if the MYSM1-ER interaction can be targeted using DUB inhibitors, as requested.
- Area of expertise of the reviewer - breast cancer, nuclear receptors, estrogen receptor biology, epigenetics, bioinformatics

3. How much time do you estimate the authors will need to complete the suggested revisions:

Estimated time to Complete Revisions (Required)

(Decision Recommendation)

Between 3 and 6 months

Yes

Revision Plan

Manuscript number: RC-2022-01807R

Corresponding author(s): Yue, Zhao

[The “revision plan” should delineate the revisions that authors intend to carry out in response to the points raised by the referees. It also provides the authors with the opportunity to explain their view of the paper and of the referee reports.]

The document is important for the editors of affiliate journals when they make a first decision on the transferred manuscript. It will also be useful to readers of the reprint and help them to obtain a balanced view of the paper.

*If you wish to submit a full revision, please use our "Full Revision" template. **It is important to use the appropriate template to clearly inform the editors of your intentions.**]*

General Statements [optional]

We really appreciate the constructive comments and suggestions of the peer-reviewers. According to their comments, we have revised our manuscript and addressed all the comments raised by the reviewers using the template of the revision plan. We plan to perform additional experiments and analyses to complete the full revision. We hope that the substantially revised version of our manuscript will be accepted by affiliate journals.

In order to conveniently communicate with each other, would you please contact us with the following email address: **yzhao30@cmu.edu.cn**

Description of the planned revisions

Reviewer #1 (Evidence, reproducibility and clarity (Required)):

Summary:

ER+ breast cancer is the most common form of cancer. Targeting ER-alpha transcriptional cofactors present one potential method to target the disease. The authors demonstrate that MYSM1 is a histone deubiquitinase and a novel ER cofactor, functioning by up-regulating ER action via histone deubiquitination.

Loss of MYSM1 attenuated cell growth and increase breast cancer cell lines' sensitivity to anti-estrogens. The authors, therefore, propose MYSM1 as a potential therapeutic target for endocrine resistance in Breast cancer.

Major Comments:

The data as presented is convincing, and the evidence for the role of MYSM1 as a co-activator of ER-alpha is extensive. Given the amount of data, I do not believe any additional experiments are needed. I could not find any description of ethics for the patient samples used.

Revision Plan

Response: Appreciate for the positive response from the reviewer. According to the important suggestions, the ethics approval for the patient specimens have been included in the “Materials and methods” part.

Minor Comments:

(1)- Figure 4C - is the increase of binding in response to Estrogen significant? It is an important control to show for MCF7 as Fig 4B is in T47D.

Response: According to the comments from reviewers, we conducted statistical difference analysis in Figure 4C, our results have shown that the recruitment of MYSM1 or ER α on c-Myc ERE region is significantly increased upon E2 treatment in MCF-7 cells.

(2)- Figure 6 - Can we clarify that B = Before, A = After

Response: Apologize for the unclear description in Figure 6. As clarified by the reviewer, “B” represents before AI treatment, “A” represents after AI treatment. We have included the description in the “Figure legends” section.

(3)- The use of Fig EV was confusing to me, I assume it means supplementary?

Response: Thank you for your question. Since our priority affiliate journal is belong to EMBO Press, this manuscript was written according to the relevant requirements and “EV” is the abbreviation of “Expanded View”, which is the same as that of the supplementary figures.

Reviewer #1 (Significance (Required)):

- Discovery science to understand the regulation of the ER is critical in discovering new opportunities to target breast cancer. As far as I can tell this is the first study where MYSM1 is a co-regulator of the ER.

- The significance would be greatly increased if the manuscript identified opportunities to target the ER via this pathway using existing compounds. However, it is reasonable to consider this is beyond the scope of this study.

Response: According to your valuable suggestion, we thus turned to screen the commercially-available compound in ZINC database to find the compounds that could spatially interact with MYSM1 protein, thereby inhibiting the activity of MYSM1. We plan to perform the additional biological function experiments to explore the effect of MYSM1-targeting compounds on the sensitivity of breast cell lines to anti-estrogen treatment.

Reviewer #2 (Evidence, reproducibility and clarity (Required)):

Below I outline a few suggestions that can help clarify specific aspects of the study.

Fig. 2: Ideally a rescue study with a wild-type and catalytically mutant MYSM1 should be performed.

Response: Thank you for your suggestion. To address this point, we will perform a rescue study with a wild-type and catalytically mutant MYSM1 in the breast cancer cells with stable knocked down of MYSM1 to examine the corresponding protein expression of ER α target genes.

Revision Plan

What is the ERa interactome in the presence and absence of MYSM1? Proteomics studies upon shMYSM1 should be performed. Alternatively, a better characterization of ERa-containing complexes upon shMYSM1 should be performed.

Response: We agree with the reviewer's suggestion to functionally address the influence of MYSM1 on ERa interactome. In breast cancer cells with the presence or absence of MYSM1, Co-IP experiments will be conducted to examine the influence of MYSM1 on the interaction between ERa and KAT2B, EP300 and CREBBP complex, which are predicted from String database.

Fig. 3: Does MYSM1 control its own protein via deubiquitination?

Response: We thank the reviewer for this suggestion and it provides us with a novel perspective upon MYSM1 investigation of whether MYSM1 is the deubiquitination substrate of itself. We would first transfect MYSM1-FL or MYSM1-ΔMPN plasmids and detect whether the endogenous MYSM1 expression changes. Next step, ubiquitination assays will be performed to determine whether MYSM1 control its own protein via deubiquitination.

Fig. 4: I propose that the authors perform MYSM1 ChIP-Seq to better show the MYSM1 distribution and overlap with ERa distribution.

Response: Appreciate for the reviewer for the valuable and important comments. ChIP-seq will be additionally performed in MCF-7 cells with MYSM1 antibody to examine the MYSM1 occupation on global chromatin in response to E2 and to show its overlap with ERa distribution.

Fig. 7. Is there a correlation between MYSM1 mRNA and protein levels in cancer and physiological samples? How is the MYSM1 transcriptionally regulated in physiological and cancer cells?

Response: We thank the reviewer for raising this issue. We will detect MYSM1 mRNA and protein levels in breast cancer and physiological samples, along with physiological and breast cancer cells. Statistics for MYSM1 transcriptional level will be further displayed.

Reviewer #2 (Significance (Required)):

This is a very comprehensive study characterizing the role of MYSM1 deubiquitinase in ERa transcriptional programs in breast cancer systems. Breast cancer therapy is an unmet need and the role of deubiquitinases warrants further investigation. This accounts for the high significance of the story.

Reviewer #3 (Evidence, reproducibility and clarity (Required)):

Luan et al performed a detailed analysis on the potential coactivator MYSM1 and its role in regulating the expression of ER and ER-dependent genes by being a deubiquitinase of ER as well the repressive mark, H2Aub1. This study has demonstrated an excellent work on the biochemistry aspect of the story with meticulous work on the role of specific domains of MYSM1 and ER and how they interact and how the deubiquitination process is regulated. This was

Revision Plan

identified initially in *Drosophila* models, but eventually and promptly explored in multiple breast cancer cell lines and patient samples.

Major Comments:

1. It is really exciting to see how MYSM1 regulates ER activity and it looks like expression of ER is the first event of regulation by MYSM1's. However, H3Ac would be the very intermediate event of ER activity. This brings a question of whether ER complex itself is affected by MYSM1 - for example, does MYSM1 affect p300, SWI/SNF and other ER-associated coactivator binding? Does it affect chromatin accessibility? Which exact histone mark of H3Ac is affected, as different proteins are involved in the acetylation of histones.

Response: Appreciate the reviewer for the valuable questions. The Co-IP and ChIP experiments will be conducted respectively to assess the influence of MYSM1 on the binding of ERα with its associated co-activators and their recruitments on EREs upon MYSM1 knockdown. In addition, ChIP assays will also be performed to determine the effect of MYSM1 on histone modification levels (H3K9ac, H3K27ac, et al). MNase assay will be further performed to examine the function of MYSM1 on chromatin accessibility.

2. The regulation of MYSM1 is mainly shown on promoters of ER regulated genes. However, ER primarily bind to enhancers. Is there any general effect on enhancers?

Response: Thank you for your comments. We will perform ChIP assays to detect the regulation of MYSM1 on ERα binding to enhancers of ERα regulated genes in breast cancer cells.

3. MYSM1 is not the complex usually cells prefer to deubiquitinate H2Aub, but BAP1. What is the role of BAP1 here? Are they redundant or any cross-talk?

Response: Concerning this interesting question, it has been reported that BAP1 co-activator function correlated with increased H3K4me3 and concomitant deubiquitination of H2Aub at target genes. However, BAP1 has not been reported as an ERα co-regulator so far. Moreover, the interaction between ERα and BAP1 cannot be predicted using the STRING database. Whether BAP1 plays a similar role as MYSM1 in breast cancer and how MYSM1 cooperates with the other DUBs to regulate the genome-wide landscape of histone H2A ubiquitination and the gene expression profiles of different mammalian cell types remains to be elusive. It would be necessary to further study in the future.

4. Effect of MYSM1 on histone marks on the EREs - only one ERE is shown. Multiple EREs should be validated by qPCR. Enhancers should also be focused. Does it affect H3K27ac or H3K4me1?

Response: Thank you for your suggestions. ChIP experiments will be conducted to examine the effect of MYSM1 on histone marks on multiple EREs of ERα target genes. Furthermore, we will focus on the effect on MYSM1 on histone marks (H3K27ac and H3K4me1 levels, et al) on enhancers of ERα target genes.

5. It is clear that MYSM1 is required for the response to antiestrogen therapies. However, the link to resistance is not completely clear. This should be investigated with multiple Tamoxifen

Revision Plan

resistant cell lines. There is one cell line used, but it is responding to tamoxifen even at lower concentrations in Crystal violet assays. MYSM1 overexpression in nonresponders doesn't mean that their activity is also more. Binding analyses should be analysed in proper Tamoxifen-resistant cell lines. Usually, Tamoxifen is used or works at concentrations from 100 nM - 1 uM in vitro to see the transcriptional effects. However, the authors claim that these are very high concentrations, but actually they aren't the concentrations which promote toxicity.

Response: We thank reviewer for the valuable comments. According to your suggestion, we will construct Tamoxifen-resistant MCF-7 or T47D cell lines carrying stable knockdown of MYSM1 to perform the biological function experiments with appropriate Tamoxifen concentrations to further confirm the effect of MYSM1 on the sensitivity of cells to anti-estrogen. In addition, we will examine the expression of MYSM1 and ER α target genes or histone H2Aub levels in nonresponders samples to preliminarily determine the activity of MYSM1 in AI-resistant samples.

6. Discussion about DUB inhibitors - how specific are these? Would they be useful to target MYSM1 activity and thus ER regulation in nonresponders or resistant cell lines? This would add up strongly on the clinical potential of the study.

Response: The DUB inhibitors mentioned in discussion are specific to USP14 and UCHL5, but not MYSM1. We thus turned to screen the commercially-available compound in ZINC database to find the compounds that could spatially interact with MYSM1 protein, thereby inhibiting the activity of MYSM1. We plan to perform the additional biological function experiments to explore the effect of MYSM1-targeting compounds on the sensitivity of breast cell lines to anti-estrogen treatment.

7. OPTIONAL: ChIP-seq analyses on the factors would be more informative to look at the unbiased mechanisms including enhancers.

Response: We appreciate your important comments. We plan to perform ChIP-seq in MCF-7 cells with MYSM1 antibody to examine the MYSM1 occupation on global chromatin in response to E2 and to show its overlap with ER α distribution.

8. Number of replicates aren't clear in figure legends. Are they biological or technical replicates?

Response: We thank for your comments. We have included the number of replicates in the "materials and methods" and "Figure legends" sections.

Minor comments:

9. Please give page numbers and line numbers in the manuscript.

Response: We have given page numbers and line numbers in the revised manuscript.

10. Title - "MYSM1 co-activates ER action". "Action" is not needed to be mentioned here.

Response: We have modified the title according to the reviewer's suggestion. The title has been modified as below: "MYSM1 acts as a novel co-activator of ER α via histone and non-histone deubiquitination to confer antiestrogen resistance in breast cancer".

Revision Plan

11. Abstract talks about the work on *Drosophila* mainly, but apart from the first experiment, everything else is done on mammalian cell culture and also clinically relevant patient samples.

Response: Thank you for your important comments. We have modified the abstract contents with breast cancer-derived cell lines instead of *Drosophila* experimental system.

12. Abstract Line 13 - the work is done many ER regulated genes and not gene.

Response: We've modified the text into "ERa-regulated genes" in Abstract section.

13. Pg 6 first paragraph - What/how many mutants were screened here?

Response: Thank you for your suggestion. In this study, about 300 fly lines carrying loss of function mutants obtained from Bloomington Stock Center were used for screening.

14. CoIP protocol is not clear. It says followed with manufacturer instructions but no kit information is provided.

Response: Apologize for the misrepresentation. Co-IP experiments were performed as that in the previous study. We have corrected the description for CoIP protocol and cited our previous study in the Materials and Methods section.

15. Fig. 1H, etc - can you show a zoomed in or DAPI removed (from merge) picture to show the interactions clearly? It's hard to follow the yellow co-interaction spots as they are hidden behind the blue colour. Any kind of quantification analyses would be wonderful.

Response: Thank you for your suggestion, we have merged the red and green colours to precisely show the co-location of MYSM1 and ERa.

16. Fig. EV1H - can you link this with the results from Fig. 1F to discuss if the delta SANT-MYSM1 lost the interaction with ER also in the IF studies?

Response: Thank you for your question. Commonly, the fluorescence intensity of confocal results mainly represents the amount of ectopic expression of MYSM1 or ERa, Co-IP experiments more exactly represent the association between proteins. It would be better to pick up the similar cell number in confocal experiments to assess the intensity of protein interaction. We will repeat the confocal again to show the exact fluorescence intensity.

17. Pg 7 - 3-4th line from last - These lines should move above where AF1 and AF2 are introduced. According to Fig. 1G, the interaction of AF2 and MYSM1 is important. Why do we see an effect on AF1 as well in Fig. 2B?

Response: Thank you for your comments. The GST ERa-AF1 and GST ERa-AF1 fusion proteins contain 29-180aa and 282-595aa of ERa truncated mutants respectively, while the ERa-AF1 and ERa-AF2 expression plasmids used in luciferase assay in Fig 2B encode 1-282aa and 178-595aa fragments. We can see the ERa-AF1 mutant in Fig 2B contains more amino acid segments than that in GST ERa-AF1 in Fig. 1G. We speculate that MYSM1 may interact with the extra segment (180aa-282aa) to upregulate ERa-AF1 induced transcription. To make it clear, we have included relevant description in the text along with a schematic representation of

Revision Plan

ERa, ERa-AF1, and ERa-AF2 plasmids used in luciferase reporter assays in Fig EV2B and in materials and methods section.

18. It's confusing to have HEK and breast cancer cell line datasets swapped inconsistently between main figures and Supplementary figures. It would be nice to keep them consistent.

Response: We have reverse the order of Fig 2B and Fig EV2C to maintain the consistency of the cell line datasets.

19. RPMI is spelled wrong in Pg. 19.

Response: We have corrected the spelling error of RPMI.

20. How long is the estrogen treatment done in each experiment? What is the concentration? This should be mentioned in the figure legends. 12 or 24 hrs time point is a later stage of estrogen receptor induction. Even 1-3 hrs would be sufficient to promote a stronger effect on RNA transcription than that of these later time points. What you are looking at is all effect on later time points and the effect should be observed on earlier time points to observe dynamic and immediate effects. p-values are required for the comparison on no E2 vs E2 here.

Response: We appreciate your valuable comment. We have rephrased the description on estrogen treatment in "Material and methods" and "Discussion" parts to more clearly state that E2 (100nM) was given for 4-6h in the experiments detecting transcriptional levels, while 16-18h in the experiments detecting translation levels. In addition, p-values have included to display the change of MYSM1 and ERa recruitment on ERE region upon E2 treatment.

21. Fig. 2G - effect on c-Myc after MYSM1 knockdown is not clear comparing to the previous WB in 2E.

Response: We will replace a clear image in Fig 2G to show the change of c-Myc protein expression after MYSM1 knockdown.

22. Pg. 8 - start of the last paragraph - "Unexpectedly, in Co-IP experiment as shown in Figure 2E and F" - These are not Co-IP experiments.

Response: Apologize for the writing error. We have re-written the sentence "Unexpectedly, in western blot experiments as shown in Figure 2E and F" in line 229.

23. Fig. 3C and E - Quantification with comparison needed.

Response: Relative ERa levels were semi-quantified by densitometry and normalized by the relative expression of 0 hour to compare the ERa degradation rate in Fig 3C and E.

24. Pg 10 - subtitle - multiple gene promoters have been looked, but the subtitle says "gene". Only ERE for c-MYC is looked at, but it says EREs.

Response: We have modified the word "genes" and "ERE" in correct forms in the text.

Revision Plan

25. MYSM1 is in the nucleus in IF even before E2 treatment, however it is recruited after estrogen treatment in ChIP assays. Explain why there is a difference seen here. What other targets they might bind to in the nucleus?

Response: The aim of ChIP experiments is to examine the recruitment of MYSM1 protein on the DNA in the presence of E2, while IF results represent the MYSM1 subcellular distribution in the nucleus even in the absence of E2. MYSM1 has been reported to bind to promoters of numerous target genes, including Ebf1 in B cell progenitors, Pax5 in naïve B cells, miR150 in B1a cells, Id2 in NK cell progenitors, Flt3 in dendritic cell precursors, and Gfi1 in hematopoietic stem and progenitor cells. In our study, we plan to perform ChIP-seq to further show its potential binding elements on the global genome in ER-positive breast cancer.

26. Pg. 10 last line - the sentence should be combined with comma.

Response: Thank you for pointing this out, we have combined a comma in the sentence.

27. Fig. 5H - What about Ki67 which is a proliferative marker for cancer cell growth?

Response: We will further perform IHC experiments to compare Ki67 expression in the shCtrl and shMYSM1 group of xenograft tumors from nude mice.

28. Pg 12 - Samples were used from patients treated with AI adjuvant treatment. A small summary of details are needed here including n, arm, details of administration, etc even though mentioned in Methods.

Response: We have restated the patients' condition and administration details in lines 342-250.

29. MYSM1 is upregulated in nonresponders, but it is also downregulated in responders which is ignored. What would this mean mechanistically? Don't patients need MYSM1 for the response or after treatment? Does estrogen inhibition regulate MYSM1 upstream?

Response: Appreciate for your important questions. The changes of intracellular environment caused by AI treatment are complicated and varied. The mechanism underlying such a phenomenon is largely unclear. We plan to perform western blot and ubiquitination assays to compare the expression and activity of MYSM1 in endocrine-resistant breast cancer cells treated or untreated with endocrine drugs to identify the effects of estrogen inhibition on MYSM1 expression. Moreover, we will detect whether MYSM1 expression is correlated with cell cycle and cell proliferation states.

30. Pg 13 - Is this data associated with any trial? More details are needed.

Response: We appreciate for your helpful comment. We have rearranged the logic of the article in order to clarify our reasoning for presenting this data. The modified contents included in lines 367-371 in the modified version are followed below: The regulation of MYSM1 on ER α action indicates that MYSM1 acts as a novel ER α co-activator, suggesting that MYSM1 may play an important role in breast cancer. We then conducted western blot and IHC experiments to estimate MYSM1 expression and the correlation between MYSM1 expression and clinicopathologic factors of the patients.

Revision Plan

31. Last lines of Pg 15 - These were already introduced in the results.

Response: We thank reviewer for their highlighting this redundancy in our text. We have simplified the text in lines 442-444.

32. Pg 16 - third last line of the first paragraph - makes typo.

Response: Thanks for pointing out this typo. We have corrected the word "make" in line 456.

Reviewer #3 (Significance (Required)):

Significance

- The study seems to be novel as MYSM1 is never studied before as a coactivator for ER. This expands the wealth of knowledge we have on coactivators which can be explored for its potential targeting to treat advanced breast cancers. The study seems to support the biochemical aspects of ER interaction, but vaguely uncovers the functional or epigenetic mechanisms.
- Studies on coactivators/coregulators of ER is very important, as modulating ER alone is not efficient enough to solve the puzzle of antiestrogen resistance. The expression/activity levels of the coregulators are very important as these can be modulated in cancers due to epigenetic reprogramming during resistance and mutations on these genes dominate. They can also serve as potential targets especially when cells don't respond to classical ER targeting therapies.
- Strength - Biochemical analyses of the interactions and detailed mechanistic information
- Limitation - Studies are very much limited to the biochemical regulation on ER and not on the molecular or epigenetic mechanisms. Association of MYSM1 in resistance mechanisms isn't clear.
- Audience - this can be interesting for both basic research and clinical audience. Biochemical knowledge would help people to understand how a nonclassical deubiquitinase can promote nuclear receptor associated transcription by targeting genomic and nongenomic targets simultaneously. Clinically this study would be relevant if the MYSM1-ER interaction can be targeted using DUB inhibitors, as requested.
- Area of expertise of the reviewer - breast cancer, nuclear receptors, estrogen receptor biology, epigenetics, bioinformatics

Description of the revisions that have already been incorporated in the transferred manuscript

Please insert a point-by-point reply describing the revisions that were already carried out and included in the transferred manuscript. If no revisions have been carried out yet, please leave this section empty.

Revision Plan

Description of analyses that authors prefer not to carry out

Please include a point-by-point response explaining why some of the requested data or additional analyses might not be necessary or cannot be provided within the scope of a revision. This can be due to time or resource limitations or in case of disagreement about the necessity of such additional data given the scope of the study. Please leave empty if not applicable.

8th Mar 2023

Dear Prof. Zhao,

Thank you for the submission of your manuscript to our editorial offices. I have now had the opportunity to read it, together with the referees' reports and your rebuttal letter, and to discuss them with the other members of our editorial team.

We agree that the study fits the scope of the journal, and we appreciate that you are willing to address/have addressed the points raised by the reviewers. We thus encourage you to submit a revised version of your manuscript, including the modifications and revisions described in your point-by-point letter. In particular, we would appreciate pharmacological targeting of MYSM1, which should reinforce the translational impact of your findings.

Acceptance of the manuscript will entail a second round of review. EMBO Molecular Medicine encourages a single round of revision only and therefore, acceptance or rejection of the manuscript will depend on the completeness of your responses included in the next, final version of the manuscript. For this reason, and to save you from any frustrations in the end, I would strongly advise against returning an incomplete revision.

When submitting your revised manuscript, please carefully review the instructions that follow below. Failure to include requested items will delay the evaluation of your revision:

We require:

4) A .docx formatted letter INCLUDING the reviewers' reports and your detailed point-by-point responses to their comments. As part of the EMBO Press transparent editorial process, the point-by-point response is part of the Review Process File (RPF), which will be published alongside your paper.

5) A complete author checklist, which you can download from our author guidelines (<https://www.embopress.org/page/journal/17574684/authorguide#submissionofrevisions>). Please insert information in the checklist that is also reflected in the manuscript. The completed author checklist will also be part of the RPF.

6) Please note that all corresponding authors are required to supply an ORCID ID for their name upon submission of a revised manuscript.

7) It is mandatory to include a 'Data Availability' section after the Materials and Methods. Before submitting your revision, primary datasets produced in this study need to be deposited in an appropriate public database, and the accession numbers and database listed under 'Data Availability'. Please remember to provide a reviewer password if the datasets are not yet public (see <https://www.embopress.org/page/journal/17574684/authorguide#dataavailability>).

8) For data quantification: please specify the name of the statistical test used to generate error bars and P values, the number (n) of independent experiments (specify technical or biological replicates) underlying each data point and the test used to calculate p-values in each figure legend. The figure legends should contain a basic description of n, P and the test applied. Graphs must include a description of the bars and the error bars (s.d., s.e.m.). Please provide exact p values.

9) Our journal encourages inclusion of *data citations in the reference list* to directly cite datasets that were re-used and

obtained from public databases. Data citations in the article text are distinct from normal bibliographical citations and should directly link to the database records from which the data can be accessed. In the main text, data citations are formatted as follows: "Data ref: Smith et al, 2001" or "Data ref: NCBI Sequence Read Archive PRJNA342805, 2017". In the Reference list, data citations must be labeled with "[DATASET]". A data reference must provide the database name, accession number/identifiers and a resolvable link to the landing page from which the data can be accessed at the end of the reference. Further instructions are available at .

13) Author contributions: CRediT has replaced the traditional author contributions section because it offers a systematic machine readable author contributions format that allows for more effective research assessment. Please remove the Authors Contributions from the manuscript and use the free text boxes beneath each contributing author's name in our system to add specific details on the author's contribution. More information is available in our guide to authors.

16) As part of the EMBO Publications transparent editorial process initiative (see our Editorial at <http://embomolmed.embopress.org/content/2/9/329>), EMBO Molecular Medicine will publish online a Review Process File (RPF) to accompany accepted manuscripts.

In the event of acceptance, this file will be published in conjunction with your paper and will include the anonymous referee reports, your point-by-point response and all pertinent correspondence relating to the manuscript. Let us know whether you agree with the publication of the RPF and as here, if you want to remove or not any figures from it prior to publication. Please note that the Authors checklist will be published at the end of the RPF.

I look forward to receiving your revised manuscript.

Yours sincerely,

Lise Roth

Rev_Com_number: RC-2022-01807

New_manu_number: EMM-2023-17672

Corr_author: Zhao

Title: MYSM1 acts as a co-activator of ER α to confer antiestrogen resistance in breast cancer

Full Revision

Manuscript number: RC-2022-01807R

Corresponding author(s): Yue, Zhao

[Please use this template only if the submitted manuscript should be considered by the affiliate journal as a full revision in response to the points raised by the reviewers.]

*If you wish to submit a preliminary revision with a revision plan, please use our "Revision Plan" template. **It is important to use the appropriate template to clearly inform the editors of your intentions.**]*

1. General Statements [optional]

We really appreciate the constructive comments and suggestions of the peer-reviewers. According to their comments, we have revised our manuscript and addressed all the comments raised by the reviewers. We have performed additional experiments and analyses to complete this full revision. We hope that the substantially revised version of our manuscript will be accepted by *EMBO Molecular Medicine*.

Reviewer #1 (Evidence, reproducibility and clarity (Required)):

Summary:

ER+ breast cancer is the most common form of cancer. Targeting ER-alpha transcriptional cofactors present one potential method to target the disease. The authors demonstrate that MYSM1 is a histone deubiquitinase and a novel ER cofactor, functioning by up-regulating ER action via histone deubiquitination.

Loss of MYSM1 attenuated cell growth and increase breast cancer cell lines' sensitivity to anti-estrogens. The authors, therefore, propose MYSM1 as a potential therapeutic target for endocrine resistance in Breast cancer.

Major Comments:

The data as presented is convincing, and the evidence for the role of MYSM1 as a co-activator of ER-alpha is extensive. Given the amount of data, I do not believe any additional experiments are needed. I could not find any description of ethics for the patient samples used.

Response: Appreciate for the positive response from the reviewer. According to the important suggestions, the ethics approval for the patient specimens have been included in the “Materials and methods” part.

Minor Comments:

(1)- Figure 4C - is the increase of binding in response to Estrogen significant? It is an important control to show for MCF7 as Fig 4B is in T47D.

Response: According to the comments from reviewers, we conducted statistical difference analysis in new Figure 4D (original panel Figure 4C), our results have shown that the recruitment of MYSM1 or ERα on c-Myc ERE region is significantly increased upon E2 treatment in MCF-7 cells.

(2)- Figure 6 - Can we clarify that B = Before, A = After

Response: Apologize for the unclear description in new Figure 7 (original panel Figure 6). As clarified by the reviewer, “B” represents before AI treatment, “A” represents after AI treatment. We have included the description in the “Figure legends” section.

(3)- The use of Fig EV was confusing to me, I assume it means supplementary?

Response: Thank you for your question. Since our priority affiliate journal is belong to EMBO Press, this manuscript was written according to the relevant requirements and “EV” is the abbreviation of “Expanded View”, which is the same as that of the supplementary figures.

Reviewer #1 (Significance (Required)):

- Discovery science to understand the regulation of the ER is critical in discovering new opportunities to target breast cancer. As far as I can tell this is the first study where MYSM1 is a co-regulator of the ER.

- The significance would be greatly increased if the manuscript identified opportunities to target the ER via this pathway using existing compounds. However, it is reasonable to consider this is beyond the scope of this study.

Response: According to your valuable suggestion, we thus turned to screen the commercially-available compound in ZINC database and found Imatinib, the compound that could spatially interact with MYSM1 protein, thereby inhibiting the activity of MYSM1. We also performed several additional biological function experiments and verified the effect of this MYSM1-targeting molecule on the sensitivity of breast cell lines to anti-estrogen treatment, as shown in new Figure 6J-K and Figure EV6G-M..

Reviewer #2 (Evidence, reproducibility and clarity (Required)):

Below I outline a few suggestions that can help clarify specific aspects of the study.

Fig. 2: Ideally a rescue study with a wild-type and catalytically mutant MYSM1 should be performed.

Response: Thank you for your suggestion. To address this point, we performed a rescue study with a wild-type and catalytically mutant MYSM1 in the breast cancer cells with stable knocked down of MYSM1 to examine the corresponding protein expression of ER α target genes. Ectopic expression of MYSM1 increased c-Myc, CCND1 and VEGF protein levels, while the catalytically mutant MYSM1 hardly changed the expression of these proteins, indicating the necessity of MYSM1 enzyme activity on ER α activity (new Figure 2H).

What is the ER α interactome in the presence and absence of MYSM1? Proteomics studies upon shMYSM1 should be performed. Alternatively, a better characterization of ER α -containing complexes upon shMYSM1 should be performed.

Response: We appreciate for your important comments. To functionally address the influence of MYSM1 on ER α interactome, in breast cancer cells with the presence or absence of MYSM1, Co-IP experiments have been conducted to examine the influence of MYSM1 on the interaction between ER α and KAT2B, EP300 and CREBBP complex, which are predicted from String database. The results showed that the association between HATs co-regulatory complex and ER α was diminished with MYSM1 knockdown, suggesting that MYSM1 is required for the associate

between HATs complex and ER α (new Fig 4C). In addition, ChIP assay results showed that MYSM1 facilitated the recruitment of ER α -containing complexes to cis regulatory elements of ER α target genes (new Fig 4D).

Fig. 3: Does MYSM1 control its own protein via deubiquitination?

Response: We thank the reviewer for this suggestion and it provides us with a novel perspective upon MYSM1 investigation of whether MYSM1 is the deubiquitination substrate of itself. We transfected MYSM1-FL or MYSM1- Δ MPN plasmids carrying FLAG tag and detected the endogenous or exogenous MYSM1 expression changes. There was no significant changes in endogenous or exogenous MYSM1 (MYSM1-HA) expression, indicating that MYSM1 may have no effect on its own protein expression (new Fig EV3).

Fig. 4: I propose that the authors perform MYSM1 ChIP-Seq to better show the MYSM1 distribution and overlap with ER α distribution.

Response: Appreciate for the reviewer for the valuable and important comments. ChIP-seq has been additionally performed in MCF-7 cells with MYSM1 antibody to examine the MYSM1 distribution on global chromatin in response to E2. ChIP-seq analysis indicate that the half-EREs are the significant enrichment sites for MYSM1 on ER α -regulated genes (new Fig 5B). To gain the functional insight into MYSM1 binding targets, Gene Oncology analyses for MYSM1-distribution-on-promoter genes were performed. Genes in E2-dependent group were highly enriched for biological process (BP) ontology terms “translation initiation”, “structural constituent of ribosome”, “mitotic cell cycle”, and “positive regulation of DNA-templated transcription, elongation” (new Fig 5D). And MYSM1 overlaps with ER α distribution on *CCND1* and *c-Myc* gene regions are displayed in new Figure 5E and 5F.

Fig. 7. Is there a correlation between MYSM1 mRNA and protein levels in cancer and physiological samples? How is the MYSM1 transcriptionally regulated in physiological and cancer cells?

Response: We thank the reviewer for raising this issue. We have detected MYSM1 mRNA and protein levels in breast cancer and physiological samples and the statistics for MYSM1 transcriptional level has been shown in new Figure 8.

Reviewer #2 (Significance (Required)):

This is a very comprehensive study characterizing the role of MYSM1 deubiquitinase in ER α transcriptional programs in breast cancer systems. Breast cancer therapy is an unmet need and the role of deubiquitinases warrants further investigation. This accounts for the high significance of the story.

Reviewer #3 (Evidence, reproducibility and clarity (Required)):

Luan et al performed a detailed analysis on the potential coactivator MYSM1 and its role in regulating the expression of ER and ER-dependent genes by being a deubiquitinase of ER as well the repressive mark, H2Aub1. This study has demonstrated an excellent work on the biochemistry aspect of the story with meticulous work on the role of specific domains of MYSM1 and ER and how they interact and how the deubiquitination process is regulated. This was identified initially in Drosophila models, but eventually and promptly explored in multiple breast cancer cell lines and patient samples.

Major Comments:

1. It is really exciting to see how MYSM1 regulates ER activity and it looks like expression of ER is the first event of regulation by MYSM1's. However, H3Ac would be the very intermediate event of ER activity. This brings a question of whether ER complex itself is affected by MYSM1 - for example, does MYSM1 affect p300, SWI/SNF and other ER-associated coactivator binding? Does it affect chromatin accessibility? Which exact histone mark of H3Ac is affected, as different proteins are involved in the acetylation of histones.

Response: Appreciate the reviewer for the valuable questions. The Co-IP and ChIP experiments have been conducted respectively to assess the influence of MYSM1 on the binding of ER α with its associated co-activators and their recruitments on EREs upon MYSM1 knockdown in new Figure 4. MYSM1 depletion attenuated the binding of ER α with the histone acetyltransferases (HATs) subunits p300/CBP/pCAF and their recruitment to the estrogen response elements (EREs) of ER α target genes. Silencing MYSM1 induced reduction of histone H3K9ac and H3K27ac levels at cis regulatory elements on promoter of ER α -regulated genes. In addition,

MNase assay was further performed and the results showed that MYSM1 may facilitate chromatin decondensation to regulate a transcription program.

2. The regulation of MYSM1 is mainly shown on promoters of ER regulated genes. However, ER primarily bind to enhancers. Is there any general effect on enhancers?

Response: Thank you for your comments. We have performed additional ChIP assays to detect the influence of MYSM1 on ER α binding to enhancers of the ER α -regulated genes, including *c-Myc* and *CCND1* in MCF-7 cells. The results have shown that MYSM1 acts as a functional co-activator of ER α , facilitating the recruitment of ER α and its co-regulatory complex to ER α -regulated enhancers (new Fig EV5E-F). Moreover, MYSM1 depletion down-regulates enhancer RNA transcription of ER α -regulated genes (new Fig EV5G).

3. MYSM1 is not the complex usually cells prefer to deubiquitinate H2Aub, but BAP1. What is the role of BAP1 here? Are they redundant or any cross-talk?

Response: Concerning this valuable question, it has been reported that BAP1 co-activator function correlated with increased H3K4me3 and concomitant deubiquitination of H2Aub at target genes. However, BAP1 has not been reported as an ER α co-regulator so far. Moreover, the interaction between ER α and BAP1 cannot be predicted using the STRING database. Whether BAP1 plays a similar role as MYSM1 in breast cancer and how MYSM1 cooperates with the other DUBs to regulate the genome-wide landscape of histone H2A ubiquitination and the gene expression profiles of different mammalian cell types remains to be elusive. It would be necessary to further study in the future.

4. Effect of MYSM1 on histone marks on the EREs - only one ERE is shown. Multiple EREs should be validated by qPCR. Enhancers should also be focused. Does it affect H3K27ac or H3K4me1?

Response: Thank you for your important suggestions. ChIP assay experiments have been further conducted to examine the regulation of MYSM1 on histone marks on multiple EREs of ER α target genes as shown in new Fig 4 and new Fig EV4. Furthermore, the levels of histone marks (H3K27ac and H3K4me1) on enhancers of ER α target genes also reduced upon MYSM1 knockdown (new Fig EV5E-F).

5. It is clear that MYSM1 is required for the response to antiestrogen therapies. However, the link to resistance is not completely clear. This should be investigated with multiple Tamoxifen resistant cell lines. There is one cell line used, but it is responding to tamoxifen even at lower concentrations in Crystal violet assays. MYSM1 overexpression in nonresponders doesn't mean that their activity is also more. Binding analyses should be analysed in proper Tamoxifen-resistant cell lines. Usually, Tamoxifen is used or works at concentrations from 100 nM - 1 uM in vitro to see the transcriptional effects. However, the authors claim that these are very high concentrations, but actually they aren't the concentrations which promote toxicity.

Response: We thank reviewer for the valuable comments. According to your suggestion, we used Tamoxifen-resistant MCF-7 or T47D cell lines carrying stable knockdown of MYSM1 to perform the biological function experiments with appropriate Tamoxifen concentrations to further confirm the effect of MYSM1 on the sensitivity of breast cancer cells to anti-estrogen treatment. Our results demonstrated that MYSM1 depletion enhances the sensitivity of Tamoxifen-resistant MCF-7 or T47D cell lines to antiestrogen treatment (new Fig7 and Fig EV7).

6. Discussion about DUB inhibitors - how specific are these? Would they be useful to target MYSM1 activity and thus ER regulation in nonresponders or resistant cell lines? This would add up strongly on the clinical potential of the study.

Response: The DUB inhibitors mentioned in discussion are specific to USP14 and UCHL5, but not MYSM1. We thus turned to screen the commercially-available compound in ZINC database to find the compounds that could spatially interact with MYSM1 protein, thereby inhibiting the activity of MYSM1. Imatinib was selected out and additional biological function experiments were performed with this small molecule to explore the effect of MYSM1-targeting compounds on the sensitivity of breast cell lines to anti-estrogen treatment. The results suggest that Imatinib as a MYSM1 inhibitor is involved in the reverse of Tamoxifen resistance (Fig 6J-K and Fig EV6G-M).

7. OPTIONAL: ChIP-seq analyses on the factors would be more informative to look at the unbiased mechanisms including enhancers.

Response: We appreciate your important comments. We have performed ChIP-seq in MCF-7 cells with MYSM1 antibody to examine the MYSM1 occupation on global chromatin in response to E2. ChIP seq analysis indicate that MYSM1 recruitment probably induce a new transcriptional landscape in the presence of estrogen. Motif analysis of MYSM1 cistrome showed an expected enrichment of ER α (half-EREs). In addition, NF-kB, GATA1, or TBP (TATA-box binding protein) binding sites are significant enrichment sites for MYSM1 on estrogen-induced genes (new Fig 5B). The corresponding contents are included in new result section in the revised version of manuscript.

8. Number of replicates aren't clear in figure legends. Are they biological or technical replicates?

Response: We thank for your kind comments. We have included the number of replicates in the new “materials and methods” and “Figure legends” sections.

Minor comments:

9. Please give page numbers and line numbers in the manuscript.

Response: Thank you for your kind suggestion. We have given page numbers and line numbers in the revised manuscript.

10. Title - "MYSM1 co-activates ER action". "Action" is not needed to be mentioned here.

Response: We have modified the title according to the reviewer’s suggestion. The title has been modified as below: “MYSM1 acts as a novel co-activator of ER α via histone and non-histone deubiquitination to confer antiestrogen resistance in breast cancer”.

11. Abstract talks about the work on Drosophila mainly, but apart from the first experiment, everything else is done on mammalian cell culture and also clinically relevant patient samples.

Response: Thank you for your important comments. We have modified the abstract contents with breast cancer-derived cell lines instead of *Drosophila* experimental system.

12. Abstract Line 13 - the work is done many ER regulated genes and not gene.

Response: We've modified the text into “ER α -regulated genes” in Abstract section.

13. Pg 6 first paragraph - What/how many mutants were screened here?

Response: Thank you for your suggestion. In this study, about 300 fly lines carrying loss of function mutants obtained from Bloomington Stock Center were used for screening.

14. CoIP protocol is not clear. It says followed with manufacturer instructions but no kit information is provided.

Response: Apologize for the misrepresentation. Co-IP experiments were performed as that in the previous study. We have corrected the description for CoIP protocol and cited our previous study in the Materials and Methods section.

15. Fig. 1H, etc - can you show a zoomed in or DAPI removed (from merge) picture to show the interactions clearly? It's hard to follow the yellow co-interaction spots as they are hidden behind the blue colour. Any kind of quantification analyses would be wonderful.

Response: Thank you for your suggestion, we have merged the red and green colours to precisely show the co-location of MYSM1 and ERa.

16. Fig. EV1H - can you link this with the results from Fig. 1F to discuss if the delta SANT-MYSM1 lost the interaction with ER also in the IF studies?

Response: Thank you for your question. Commonly, the fluorescence intensity of confocal results mainly represents the amount of ectopic expression of MYSM1 or ERa, Co-IP experiments more exactly represent the association between proteins. It would be better to pick up the similar cell number in confocal experiments to assess the intensity of protein interaction.

17. Pg 7 - 3-4th line from last - These lines should move above where AF1 and AF2 are introduced. According to Fig. 1G, the interaction of AF2 and MYSM1 is important. Why do we see an effect on AF1 as well in Fig. 2B?

Response: Thank you for your comments. The GST ERa-AF1 and GST ERa-AF1 fusion proteins contain 29-180aa and 282-595aa of ERa truncated mutants respectively, while the ERa-AF1 and ERa-AF2 expression plasmids used in luciferase assay in Fig 2B encode 1-282aa and 178-595aa fragments. We can see the ERa-AF1 mutant in Fig 2B contains more amino acid segments than that in GST ERa-AF1 in Fig. 1G. We speculate that MYSM1 may interact with the extra

segment (180aa-282aa) to upregulate ERa-AF1 induced transcription. To make it clear, we have included relevant description in the text along with a schematic representation of ERa, ERa-AF1, and ERa-AF2 plasmids used in luciferase reporter assays in Fig EV2B and in materials and methods section.

18. It's confusing to have HEK and breast cancer cell line datasets swapped inconsistently between main figures and Supplementary figures. It would be nice to keep them consistent.

Response: We have reverse the order of Fig 2B and Fig EV2C to maintain the consistency of the cell line datasets.

19. RPMI is spelled wrong in Pg. 19.

Response: We have corrected the spelling error of RPMI.

20. How long is the estrogen treatment done in each experiment? What is the concentration? This should be mentioned in the figure legends. 12 or 24 hrs time point is a later stage of estrogen receptor induction. Even 1-3 hrs would be sufficient to promote a stronger effect on RNA transcription than that of these later time points. What you are looking at is all effect on later time points and the effect should be observed on earlier time points to observe dynamic and immediate effects. p-values are required for the comparison on no E2 vs E2 here.

Response: We appreciate your valuable comment. We have rephrased the description on estrogen treatment in “Material and methods” and “Discussion” parts to more clearly state that E2 (100nM) was given for 4-6h in the experiments detecting transcriptional levels, while 16-18h in the experiments detecting translation levels. In addition, *p*-values have included to display the change of MYSM1 and ERa recruitment on ERE region upon E2 treatment.

21. Fig. 2G - effect on c-Myc after MYSM1 knockdown is not clear comparing to the previous WB in 2E.

Response: We have replaced a clear image in Fig 2G to show the change of c-Myc protein expression after MYSM1 knockdown.

Full Revision

22. Pg. 8 - start of the last paragraph - "Unexpectedly, in Co-IP experiment as shown in Figure 2E and F" - These are not Co-IP experiments.

Response: Apologize for the writing error. We have re-written the sentence "Unexpectedly, in western blot experiments as shown in Figure 2E and F" in line 229.

23. Fig. 3C and E - Quantification with comparison needed.

Response: Relative ERa levels were semi-quantified by densitometry and normalized by the relative expression of 0 hour to compare the ERa degradation rate in new Fig 3C and E.

24. Pg 10 - subtitle - multiple gene promoters have been looked, but the subtitle says "gene". Only ERE for c-MYC is looked at, but it says EREs.

Response: Thank you for your kind comments. We have modified the word "genes" and "ERE" in correct forms in the text.

25. MYSM1 is in the nucleus in IF even before E2 treatment, however it is recruited after estrogen treatment in ChIP assays. Explain why there is a difference seen here. What other targets they might bind to in the nucleus?

Response: The aim of ChIP experiments is to examine the recruitment of MYSM1 protein on the DNA in the presence of E2, while IF results represent the MYSM1 subcellular distribution in the nucleus even in the absence of E2. MYSM1 has been reported to bind to promoters of numerous target genes, including Ebf1 in B cell progenitors, Pax5 in naïve B cells, miR150 in B1a cells, Id2 in NK cell progenitors, Flt3 in dendritic cell precursors, and Gfi1 in hematopoietic stem and progenitor cells. In the additional experiments, ChIP-seq was performed in revision. Motif analysis of MYSM1 cistrome showed an expected enrichment of ERa (half-EREs). In addition, NF-kB, GATA1, or TBP (TATA-box binding protein) binding sites are significant enrichment sites for MYSM1 on estrogen-induced genes (new Fig 5B).

26. Pg. 10 last line - the sentence should be combined with comma.

Response: Thank you for pointing this out, we have combined a comma in the sentence.

27. Fig. 5H - What about Ki67 which is a proliferative marker for cancer cell growth?

Response: We have further performed IHC experiments to compare Ki67 expression in the shCtrl and shMYSM1 group of xenograft tumors from nude mice as shown in new Fig EV6F.

28. *Pg 12 - Samples were used from patients treated with AI adjuvant treatment. A small summary of details are needed here including n, arm, details of administration, etc even though mentioned in Methods.*

Response: Appreciate for your important comments. We have restated the patients' condition and administration details at page 16-17 in the revised manuscript.

29. *MYSM1 is upregulated in nonresponders, but it is also downregulated in responders which is ignored. What would this mean mechanistically? Don't patients need MYSM1 for the response or after treatment? Does estrogen inhibition regulate MYSM1 upstream?*

Response: Appreciate for your important questions. The changes of intracellular environment caused by AI treatment are complicated and varied. The mechanism underlying such a phenomenon is largely unclear. Proliferation marker Ki67 is the prototypic cell cycle related nuclear protein. Ki67 index is a criterion for determining the efficacy of neoadjuvant treatment in these patients. We then explored the potential link between MYSM1 expression and cell cycle progression. Western blot experiments showed that MYSM1 gradually accumulated as the arrested cells released into cell cycle (new Fig 7B), indicating that MYSM1 protein expression might be related with the cell cycle and cell proliferation state.

30. *Pg 13 - Is this data associated with any trial? More details are needed.*

Response: We appreciate for your helpful comment. The data from western blot and IHC experiments for detecting MYSM1 expression in clinical breast cancer samples and the correlation between MYSM1 expression and clinicopathologic factors of the patients. The results from IHC staining with the anti-MYSM1 antibody demonstrated that the expression of MYSM1 gradually increased as the BCa proceeded into higher grade (new Fig 8E and F). The statistical analysis of clinicopathological characteristics proposed a tight bond in terms of MYSM1 expression with histological grade, ERa status, and HER2 status (Table 2). The details have been shown in the result section at page 18-19 in the revised manuscript.

31. Last lines of Pg 15 - These were already introduced in the results.

Response: We thank reviewer for their highlighting this redundancy in our text. We have simplified the text in lines 602-603 at page 21 in the revised version.

32. Pg 16 - third last line of the first paragraph - makes typo.

Response: Thanks for pointing out this typo. We have corrected the word “make” in line 615 at page 22 in the revised manuscript.

Reviewer #3 (Significance (Required)):

Significance

- *The study seems to be novel as MYSM1 is never studied before as a coactivator for ER. This expands the wealth of knowledge we have on coactivators which can be explored for its potential targeting to treat advanced breast cancers. The study seems to be support the biochemical aspects of ER interaction, but vaguely uncovers the functional or epigenetic mechanisms.*
- *Studies on coactivators/coregulators of ER is very important, as modulating ER alone is not efficient enough to solve the puzzle of antiestrogen resistance. The expression/activity levels of the coregulators are very important as these can be modulated in cancers due to epigenetic reprogramming during resistance and mutations on these genes dominate. They can also serve as potential targets especially when cells don't respond to classical ER targeting therapies.*
- *Strength - Biochemical analyses of the interactions and detailed mechanistic information*
- *Limitation - Studies are very much limited to the biochemical regulation on ER and not on the molecular or epigenetic mechanisms. Association of MYSM1 in resistance mechanisms isn't clear.*
- *Audience - this can be interesting for both basic research and clinical audience. Biochemical knowledge would help people to understand how a nonclassical deubiquitinase can promote nuclear receptor associated transcription by targeting genomic and nongenomic targets simultaneously. Clinically this study would be relevant if the MYSM1-ER interaction can be targeted using DUB inhibitors, as requested.*
- *Area of expertise of the reviewer - breast cancer, nuclear receptors, estrogen receptor biology, epigenetics, bioinformatics*

21st Sep 2023

Dear Prof. Zhao,

Thank you for the submission of your revised manuscript to EMBO Molecular Medicine. We have now received the reports from the referees who had originally reviewed your manuscript for Review Commons. As you will see below, they are overall satisfied with the revisions and supportive of publication pending minor revisions, and I am therefore pleased to inform you that we will be able to accept your manuscript once the following points will be addressed:

1/ Referees comments:

Please address the remaining concerns from the referees. In particular, the referees are not convinced by the new MYSM ChIP-Seq results and recommend removing these data.

2/ Manuscript text:

- Please address the queries from our data editors in the related Data Edited ms file, remove the yellow highlights and only keep in track changes mode any new modification.

- Would you consider shortening the title, for example to "MYSM1 acts as a novel co-activator of ER α to confer anti-estrogen resistance in breast cancer"?

- We can accommodate a maximum of 5 keywords, please adjust accordingly.

- Material and methods:

o Antibodies: please provide concentrations/dilutions.

o Cell culture: please indicate whether the cells were authenticated and tested for mycoplasma contamination.

o Mouse experiments: please indicate the housing and husbandry conditions. The manuscript must include a statement in the Materials and Methods identifying the institutional and/or licensing committee approving the experiments.

o Patients and tumor specimens: please include a statement that the experiments conformed to the principles set out in the WMA Declaration of Helsinki and the Department of Health and Human Services Belmont Report.

o Statistics: please provide information on randomization, blinding, sample size and exclusion/inclusion criteria.

- Data Availability: Note that the Data Availability Section is restricted to new primary data that are part of this study. Please remove "Supplementary Data are available at The EMBO Molecular Medicine online" and "This study includes no data deposited in external repositories."

- Acknowledgements: the complete funding information should be provided both in the acknowledgements and in the submission system (currently, China Postdoctoral Science Foundation (276066), Foundation of Liaoning Province of China (LJKZ0756), local projects supported by the central government (2022JH6/100100035), Foreign expert project of Ministry of Science and Technology (G2022006007L) are missing in the submission system).

- Author contributions: CRediT has replaced the traditional author contributions section because it offers a systematic machine-readable author contributions format that allows for more effective research assessment. Please remove the Authors Contributions from the manuscript and use the free text boxes beneath each contributing author's name in our system to add specific details on the author's contribution. More information is available in our guide to authors.

- Please reformat the references to have maximum 10 authors listed before et al.

3/ Figures:

- The main figures and EV figures should be uploaded as individual high resolution figure files.

- We can accommodate a maximum of 5 EV figures, please adjust accordingly. For the figures that you do NOT wish to display as Expanded View figures, they should be bundled together with their legends in a single PDF file called *Appendix*, which should start with a short Table of Content. Appendix figures should be referred to in the main text as: "Appendix Figure S1, Appendix Figure S2" etc.

- Kindly provide exact p values (including for ns, non significant) in the figures or their legends (including EV figures and Appendix).

- Please make sure that all figures and figure panels are referenced in the text. Currently, callouts are missing for Fig. 8A, B, C.

- Please add tables 1 and 2 to the manuscript file.

- Suppl. Tables 1-3 should be renamed Table EV1, EV2 and EV3 and uploaded as individual, separate files.

4/ Checklist:

- Please fill in the right column for each section where you indicated "Yes" (in which section is the information available).

- In the section "Experimental animals", you filled in "Animal observed in or captured from the field", please correct.

- In the section "Human research participants", please double-check your answer in the subsection "if collected and within the bounds of privacy constraints report on age, sex and gender or ethnicity for all study participants".

- In the statistics section, please make sure the information provided matches the information provided in the manuscript.

- Please check whether the "Dual Use Research of Concern" applies to your manuscript and adjust this section accordingly.

5/ Please note that all corresponding authors are required to supply an ORCID ID for their name upon submission of a revised

manuscript. An ORCID identified is currently missing for Prof. Qiang Zhang.

6/ The paper explained: EMBO Molecular Medicine articles are accompanied by a summary of the articles to emphasize the major findings in the paper and their medical implications for the non-specialist reader. Please provide a draft summary of your article highlighting

7/ Every published paper now includes a 'Synopsis' to further enhance discoverability. Synopses are displayed on the journal webpage and are freely accessible to all readers. They include a short stand first (maximum of 300 characters, including space) as well as 2-5 one-sentences bullet points that summarizes the paper. Please write the bullet points to summarize the key NEW findings. They should be designed to be complementary to the abstract - i.e. not repeat the same text. We encourage inclusion of key acronyms and quantitative information (maximum of 30 words / bullet point). Please use the passive voice. Please attach these in a separate file or send them by email, we will incorporate them accordingly.

8/ As part of the EMBO Publications transparent editorial process initiative (see our Editorial at <http://embomolmed.embopress.org/content/2/9/329>), EMBO Molecular Medicine will publish online a Review Process File (RPF) to accompany accepted manuscripts.

This file will be published in conjunction with your paper and will include the anonymous referee reports, your point-by-point response and all pertinent correspondence relating to the manuscript. Let us know whether you agree with the publication of the RPF and as here, if you want to remove or not any figures from it prior to publication.

I look forward to receiving your revised manuscript.

Yours sincerely,

Lise Roth

- Graphs 800-1,200 DPI
- Photos 400-800 DPI
- Colour (only CMYK) 300-400 DPI"

*Additional important information regarding figures and illustrations can be found at <https://bit.ly/EMBOPressFigurePreparationGuideline>. See also figure legend preparation guidelines: <https://www.embopress.org/page/journal/17574684/authorguide#figureformat>

**** Reviewer's comments ****

Referee #1 (Comments on Novelty/Model System for Author):

A comprehensive amount of data supports the result of a novel ER co-activator. The medical impact is high due to the identification of a medically relevant compound that can target the interaction.

Referee #1 (Remarks for Author):

The reviews addressed my concerns.

Reviewing the new data provided. Figure 5E/F raises some questions. The authors state that the top track is MYSM1 binding. However, I can't see any evidence of binding. The track looks like background noise.

The authors should include the peaks as called by MACS2 as another track for ER and MYSM1.

I was also unclear if there were three independent ChIP-seq experiments (as started for the ChIP section). If so, statistical analysis of +/- E2 on the MYMS1 signal is possible and recommended.

I cannot follow the figure legend for EV5A and 5B. There appear to be very similar data, but there is no signal in EV5B.

Referee #2 (Comments on Novelty/Model System for Author):

the models used in the manuscript are adequate.

Referee #2 (Remarks for Author):

The authors have made a significant effort and largely addressed my comments. My only comment is that the MYSM ChIP-Seq data are not convincing. The enrichment in the areas suggested a speaks is not clear. I suggest removing these data and trying ChIPing MYSM using a different protocol and maybe a different crosslinker. I think the authors can alternatively show ChIP-qPCR data for MYSM1 and ER α for the loci of interest. And show: a)shMYSM1 (+/-ESTROGEN) ChIP-qPCR data as control, b) 1-2 loci as negative control loci and c)IgG control.

Referee #3 (Comments on Novelty/Model System for Author):

Relevant models utilised. Revised manuscript shows more appropriate models to support the conclusions.

Referee #3 (Remarks for Author):

I appreciate the consideration of performing MYSM1 ChIP-seq, but the data looks more like background as evidently seen in Fig. 5E and F single gene profiles and Fig. EV5B. The peaks are everywhere and the increased and decreased binding as mentioned by the authors cant be seen at all, as the binding is everywhere. This data quality is not adequate and its better to remove this data or repeat the experiment with better antibodies or dual crosslinking. It might be worth to see if they are bound clearly on enhancers, as the current figures show only the profiles on TSS.

Rev_Com_number: RC-2022-01807

New_manu_number: EMM-2023-17672-V2

Corr_author: Zhao

Title: MYSM1 acts as a co-activator of ER α to confer antiestrogen resistance in breast cancer

Response to Reviewer's comments

Referee #1 (Comments on Novelty/Model System for Author):

A comprehensive amount of data supports the result of a novel ER co-activator. The medical impact is high due to the identification of a medically relevant compound that can target the interaction.

Referee #1 (Remarks for Author):

The reviews addressed my concerns.

Reviewing the new data provided. Figure 5E/F raises some questions. The authors state that the top track is MYSM1 binding. However, I can't see any evidence of binding. The track looks like background noise.

The authors should include the peaks as called by MACS2 as another track for ER and MYSM1.

I was also unclear if there were three independent ChIP-seq experiments (as started for the ChIP section). If so, statistical analysis of +/- E2 on the MYSM1 signal is possible and recommended.

I cannot follow the figure legend for EV5A and 5B. There appear to be very similar data, but there is no signal in EV5B.

Response: Thank you for your valuable comments, we realized that the ChIP-seq data quality is not adequate and we have to remove this data from our manuscript. We would like to perform it with dual crosslinking or better antibodies in the near future for the next project.

Referee #2 (Comments on Novelty/Model System for Author):

the models used in the manuscript are adequate.

Referee #2 (Remarks for Author):

The authors have made a significant effort and largely addressed my comments. My only comment is that the MYSM ChIP-Seq data are not convincing. The enrichment in the areas suggested a speaks is not clear. I suggest removing these data and trying ChIPing MYSM using a different protocol and maybe a different crosslinker. I think the authors can alternatively show ChIP-qPCR data for MYSM1 and ER α for the loci of interest. And show: a)shMYSM1 (+/-ESTROGEN) ChIP-qPCR data as control, b) 1-2 loci as negative control loci and c)IgG control.

Response: Thank you for your important comments and kind suggestions, we realized that the ChIP-seq data quality is not adequate and we have to remove this data from our manuscript. We would like to perform it with dual crosslinking or better antibodies in the near future for the next project. As your comments, ChIP-qPCR data for MYSM1 and ER α for the loci of interest has been shown in the Figure 4A-B.

Referee #3 (Comments on Novelty/Model System for Author):

Relevant models utilised. Revised manuscript shows more appropriate models to support the conclusions.

Referee #3 (Remarks for Author):

I appreciate the consideration of performing MYSM1 ChIP-seq, but the data looks more like background as evidently seen in Fig. 5E and F single gene profiles and Fig. EV5B. The peaks are everywhere and the increased and decreased binding as mentioned by the authors cant be seen at all, as the binding is everywhere. This data quality is not adequate and its better to remove this data or repeat the experiment with better antibodies or dual crosslinking. It might be worth to see if they are bound clearly on enhancers, as the current figures show only the profiles on TSS.

Response: We appreciate your important comments, as you mentioned, we really agree that the ChIP-seq data quality is not adequate and we have to remove this data from our manuscript. We would like to perform it with dual crosslinking or better antibodies in the near future for the next project. As your comments, ChIP-qPCR data for MYSM1 and ER α on enhances of interest has been shown in the Figure EV3A-B. The results have demonstrated that MYSM1 acting as a functional co-factor affects the binding of ER α and its co-regulatory complex to ER α -related enhancer.

6th Nov 2023

Dear Prof. Zhao,

Thank you for sending the corrected files. I am pleased to inform you that your manuscript is accepted for publication and is now being sent to our publisher to be included in the next available issue of EMBO Molecular Medicine!

Please note that I have removed "Source data are available online for this figure" in the legends of figures 1 and 2, and modified the Data information section at the end of each legend to match our format.

Please carefully check again the legends of your figures at proof stage and correct if needed.

Please also check the Statistics section where you wrote: "For all in vitro and in vivo experiments, two-tailed Student's t test was performed to calculate the P value and data in bar graphs represent mean {plus minus} SD of at least three technical replicates." Did you mean indeed biological replicates, or technical replicates? Please correct if needed at proof stage.

Yours sincerely,

Rev_Com_number: RC-2022-01807
New_manu_number: EMM-2023-17672-V3
Corr_author: Zhao
Title: MYSM1 acts as a novel co-activator of ERα to confer antiestrogen resistance in breast cancer